# An Axiomatic Framework for N-Agent Ad Hoc Teamwork: From Shapley Axioms to Learning

## Abstract

Open multi-agent systems are increasingly relevant for modelling the emerging real-world domains such as smart grids and swarm robotics. This paper addresses the recently posed problem of n-agent ad hoc teamwork (NAHT), where only a subset of agents is controllable. We propose an axiomatic game-theoretic framework for the NAHT, formulated via a cooperative game model which differentiates between the learning objectives of NAHT and MARL. Within this framework, the axiomatic characterization of the Shapley value—Efficiency, Symmetry, and Linearity—is reinterpreted as structural constraints on individual value functions. This yields a principled design space: enforcing all axioms recovers the Shapley value, while dropping Efficiency yields the Banzhaf index, leading to our Banzhaf Machine variant. As concrete instantiations, we develop Shapley Machine and Banzhaf Machine, which enforce different subsets of axioms during learning. Implemented on IPPO and POAM, these algorithms provide stronger performance.

## 1 Introduction

Multi-agent systems (MAS) have become a prominent paradigm for modelling the emerging real-world tasks such as smart grids (Wang et al., 2021), railway network management (Zhang et al., 2024), and swarm robotics (Nayak et al., 2023; Li et al., 2024). Reinforcement learning (RL) (Sutton & Barto, 2018), and in particular multi-agent reinforcement learning (MARL), has shown promise for solving such problems. However, MARL struggles under real-world conditions involving openness (where the number of uncontrolled agents can vary) and generalization (where teammate behaviours are unknown). To address these challenges, Wang et al. (2025) recently introduced the n-agent ad hoc teamwork (NAHT), which extends MARL to settings with varying and potentially unknown teammates. This paper builds on that problem, aiming to establish a theoretical foundation and a general algorithmic approach for the NAHT.

The initial practical solution to the NAHT, POAM (Wang et al., 2025), augments the IPPO algorithm (De Witt et al., 2020) with an embedding vector that represents each agent's belief about other agents' potential behaviours. Although POAM improves empirical performance, it has several key limitations: (i) It is designed heuristically without rigorous theoretical grounding, which undermines trustworthiness—a critical concern for multi-agent systems (Hammond et al., 2025). (ii) The fundamental distinction between the learning objectives of NAHT and MARL has been overlooked when designing algorithms. (iii) POAM employs TD($\lambda$) (Sutton, 1988) to train its critics, but its relation to the NAHT objective remains unclear, leaving no guarantee that the learned value functions are aligned with what NAHT fundamentally aims to address.

The Shapley value (Shapley, 1953) is a classic payoff distribution rule from cooperative game theory that has been widely applied in MARL (Wang et al., 2020; Li et al., 2021; Han et al., 2022; Wang et al., 2022; Li et al., 2023). Previously, the Shapley value's formula was used to shape rewards or value functions under the assumption that all agent are controlled, an assumption that breaks down in the NAHT setting. To investigate whether the Shapley value's axioms can provide a principled foundation for solving the NAHT, we propose a new axiomatic framework for the NAHT based on a cooperative game structure proposed by Dubey (1975) that can well describe the learning objective of the NAHT. Specifically, we build on its axiomatic characterization (Efficiency, Symmetry, and Linearity) to structure individual value functions, which lets us both recover the Shapley value (when all axioms hold) and design new variants by selectively relaxing a particular axiom.

The main contributions of this paper are as follows: (1) We cast the NAHT problem as a state-specific cooperative game, making explicit how openness alters its learning objective compared with the standard MARL. (2) We extend the Shapley value's three axioms—Efficiency, Symmetry, and Linearity—to the NAHT setting and transform them into structural constraints for designing reinforcement learning algorithms. Notably, the Linearity axiom corresponds to truncated TD($\lambda$) prediction (Cichosz, 1994). Enforcing all axioms recovers the Shapley value, while relaxing particular axioms—such as dropping Efficiency—yields alternative payoff allocations like the Banzhaf index (Banzhaf III, 1964). (3) By instantiating these constraints in learning, we propose the Shapley Machine, a generic algorithm that can be built on both IPPO and POAM, referred to as SM-IPPO and SM-POAM. In addition, when the Efficiency axiom is removed, the framework yields the Banzhaf Machine, producing the corresponding implementations BM-IPPO and BM-POAM.

We evaluate the Shapley Machine and Banzhaf Machine on modified versions of MPE-PP and SMAC adapted to the NAHT setting (Wang et al., 2025). They consistently outperform the base algorithms, and our experiments show that relaxing the Efficiency axiom may even surpass enforcing the full Shapley axioms in agent type generalization. The related work and complete mathematical proofs are left to Appendices A and D, respectively.

## 2 BACKGROUND

### 2.1 N-AGENT AD HOC TEAMWORK

In this section, we describe a problem setting for open multi-agent systems, referred to as n-agent ad hoc teamwork (NAHT) (Wang et al., 2025). The main challenges of NAHT are as follows: (1) coordination with potentially unknown types of teammates (generalization), and (2) coping with a varying number of uncontrolled teammates (openness). A decentralized partially observable Markov decision process (Dec-POMDP) (Oliehoek et al., 2016) is considered to formalize the problem. There is a team of agents $\mathcal{M}$ whose size is denoted as $M$, a state space $\mathcal{S}$, a joint action space $\mathcal{A} = \times_{i \in \mathcal{M}} \mathcal{A}_i$, a per-agent observation space $\mathcal{O}_i$, a transition function $P_T : \mathcal{S} \times \mathcal{A} \to \Delta(\mathcal{S})$, a common reward function $R_t : \mathcal{S} \times \mathcal{A} \to \mathbb{R}$, a discount factor $\gamma \in [0, 1]$ and an episode length $T \in \mathbb{Z}^+$. Each agent receives observations via an observation function $O_i : \mathcal{S} \times \mathcal{A} \to \Delta(\mathcal{O}_i)$. $\mathcal{H}_i$ is defined as an agent $i$'s space of observation and action histories. Each agent is equipped with a policy $\pi_i : \mathcal{H}_i \to \Delta(\mathcal{A}_i)$.

In this problem, there are two groups of agents: a set of controlled agents denoted as $\mathcal{C}$ whose size is denoted as $N$, and a set of uncontrolled agents denoted as $\mathcal{U}$ whose size is denoted as $M - N$. As NAHT is an open system, at the beginning of an multi-agent interaction process, a team of agents $\mathcal{M}$, consisting of $N$ agents randomly drawn from the controlled agent pool $\mathcal{C}$ and $M - N$ agents randomly drawn from the uncontrolled agent pool $\mathcal{U}$. For conciseness, each controlled agent is characterized by its policy, and the set of controlled agents can be expressed as $\mathcal{C}(\theta) = \{\pi_i^\theta\}_{i=1}^N$. The above sampling procedure is denoted as $X(\mathcal{U}, \mathcal{C}(\theta))$. The aim of NAHT is learning parameters $\theta$ to solve the following optimization problem:

$$\max_\theta \mathbb{E}_{\pi^{(\mathcal{M})} \sim X(\mathcal{U}, \mathcal{C}(\theta))} \left[ \sum_{t=0}^T \gamma^t R_t \right], \tag{1}$$

where $\pi^{(\mathcal{M})}$ denotes the joint policy of a team of agents $\mathcal{M}$. Note that different number of controlled agents $N$ would lead to various teams of agents $\mathcal{M}$.

### 2.2 REPRESENTATION OF COOPERATIVE GAMES AND THE SHAPLEY VALUE

**Definition 2.1** (**Representation of Cooperative Games** (Dubey, 1975)). *We first define such a function that $v_C^z = z$, for any $z \in \mathbb{R}$. If $C \subseteq D$, $v_C^z(D) = z$, otherwise, $v_C^z(D) = 0$. For a cooperative game described by a characteristic function $v : 2^{\mathcal{M}} \to \mathbb{R}_{\geq 0}$, we can describe a set of such games over a team of agents $\mathcal{M}$ by a set of basis games $\{v_C^1 \mid \emptyset \neq C \subseteq \mathcal{M}\}$, such as $\mathcal{G} = \{w \mid w = \sum k_C v_C^1, k_C \in \mathbb{R}, \emptyset \neq C \subseteq \mathcal{M}\}$. The analytic form of $k_C$ is represented as: $k_C = \sum_{T \subseteq C} (-1)^{|C|-|T|} v(T)$.*

**Representation of Cooperative Games.** A cooperative game is usually described as a characteristic function $v : 2^{\mathcal{M}} \to \mathbb{R}_{\geq 0}$, where $\mathcal{M}$ is a team of agents. An arbitrary cooperative game $v$ can be

uniquely represented by a set of basis games $\{v_C^1 \mid \emptyset \neq C \subseteq \mathcal{M}\}$ (Dubey, 1975), as delineated in Definition 2.1. In this paper, we focus on investigating how cooperative games are approximated over a team of agents. **Therefore, we only consider function values $v(\mathcal{M})$ and $v_C^1(\mathcal{M})$ for a team of agents $\mathcal{M}$, instead of other coalitions $C \subset \mathcal{M}$.** As per previous work in MARL (Wang et al., 2020), we consider the superadditive games as the game class of $\mathcal{G}$, which is suitable for a cooperative multi-agent task such as NAHT (see Appendix B.2 for more details). More specifically, the condition for $\mathcal{G}$ restricted to superadditive games is: $k_C \geq 0$.

**Theorem 2.2** (**Axioms of Shapley Values** (Dubey, 1975)). *Shapley value is a unique payoff allocation function on the cooperative game space $\mathcal{G}$, satisfying Efficiency, Symmetry and Additivity.*

**Shapley Values.** The Shapley value $\phi : \mathcal{G} \to \mathbb{R}^M$ is a multidimensional linear transformation defined on the set of cooperative game $\mathcal{G}$, given that there are $M$ agents in total. Each dimension of $\phi$ indicates the payoff allocation to an agent. It has been proved that Shapley value is a unique payoff allocation function on the set of cooperative games $\mathcal{G}$ which satisfies the following axioms: Efficiency, Symmetry and Additivity (Dubey, 1975),[1] highlighted in Theorem 2.2. Efficiency and Symmetry have been well investigated in the literature of multi-agent reinforcement learning (Wang et al., 2020; Li et al., 2021; Han et al., 2021; Wang et al., 2022; Chai et al., 2024), but Additivity still lacks attention. In detail, Additivity means that $\phi(w_1 + w_2) = \phi(w_1) + \phi(w_2)$, for any cooperative games $w_1, w_2 \in \mathcal{G}$. If we consider $m$ possible games, then we have the following expression: $\phi(\sum_{i=1}^m w_i) = \sum_{i=1}^m \phi(w_i)$, where $w_1, w_2, ..., w_m \in \mathcal{G}$ and the sum $\sum_{i=1}^m w_i$ can be seen as a game's value reproduced by the above games' values. Linearity is a stronger condition than Additivity (Dubey, 1975; Young, 1985). Specifically, Linearity requires $\phi(\sum_{i=1}^m \alpha_i w_i) = \sum_{i=1}^m \alpha_i \phi(w_i)$, for $\alpha_i \in \mathbb{R}$. As a special case, setting all $\alpha_i = 1$ recovers Additivity. **In this paper, we focus on the Linearity axiom to establish Additivity, which underpins the derivation of our method.**

## 3 GAME-STRUCTURED N-AGENT AD HOC TEAMWORK

To fit the dynamic environment setting in NAHT, we extend the cooperative game space $\mathcal{G}$ to the state space $\mathcal{S}$ of Dec-POMDP, forming a set of state-specific cooperative games. In more detail, for each state $s \in \mathcal{S}$ we have a state-specific cooperative game space $\mathcal{G}(s) = \{w_s \mid w_s = \sum k_C v_{C,s}^1, k_C \in \mathbb{R}, \emptyset \neq C \subseteq \mathcal{M}\}$, which is generated by a set of basis games $\{v_{C,s}^1 \mid \emptyset \neq C \subseteq \mathcal{M}\}$.

**Remark 3.1.** *Given a fixed state $s \in \mathcal{S}$, $\mathcal{G}(s)$ is isomorphic to a real coordinate space $\mathbb{R}^{2^M-1}$, consistent with the structure of $\mathcal{G}$. As a result, all properties for $\mathcal{G}$ also hold for $\mathcal{G}(s)$.*

**Definition 3.2** (**Game-Structured NAHT**). *$\mathcal{G}_{NAHT} = \times_{s \in \mathcal{S}} \mathcal{G}(s)$ denotes the set of all NAHT processes, where each element is a tuple of state-specific cooperative game values, specifying a team of agents $\mathcal{M}$, a state space $\mathcal{S}$, and weightings $k_C$ associated with basis games of all non-empty coalitions $\emptyset \neq C \subseteq \mathcal{M}$.*

By aggregating the state-specific cooperative game spaces for possible states, we represent possible NAHT processes for a team of agents as a structure $\mathcal{G}_{\text{NAHT}}$, as described in Definition 3.2. The insight is as follows: (1) Each state-specific cooperative game value captures the characteristics of a team's performance in an NAHT process initiated from the corresponding state; (2) The characteristics of state-specific cooperative game values are shaped following the principle of the cooperative game theory (see Definition 2.1) which reflects the contributing factor of every agent. Thus, ignoring any single agent would induce bias to game values; (3) A state-specific cooperative game value can be decomposed into two learning curricula, each aligned with a distinct learning sub-objective—learning **internal cooperation** among controlled agents and learning **external teamwork** with uncontrolled agents:

$$v_s = \underbrace{\sum_{C_{int} \in \mathcal{P}_+(\mathcal{C})} k_{C_{int}} v_{C_{int},s}^1}_{\textbf{Internal Cooperation}} + \underbrace{\sum_{C_{ext} \in \mathcal{P}_+(\mathcal{M}) \backslash \mathcal{P}_+(\mathcal{C})} k_{C_{ext}} v_{C_{ext},s}^1}_{\textbf{External Teamwork}},$$

where $\mathcal{P}_+(\cdot)$ indicates the powerset of a set excluding the empty set. Related to the NAHT, variation in the number of controlled agents requires learning to establish internal cooperation, whereas variation

---

[1]For an arbitrary characteristic function game value, Shapley value is uniquely determined by the Efficiency, Additivity, Symmetry, and Dummy Player axioms (Chalkiadakis et al., 2011)[Chap. 2]. Since the Dummy Player axiom is already embedded in the basis game value construction of Dubey (1975), it is automatically satisfied.

in the number and types of uncontrolled agents demands learning strategies for robust external teamwork. In contrast, the standard MARL only requires learning to establish internal cooperation among a fixed number of controlled agents with no consideration of any uncontrolled agent, so it does not necessarily require any complex structure for the learning curriculum.

**Remark 3.3.** *A state-specific cooperative game space $\mathcal{G}(s)$ is isomorphic to $\mathbb{R}^{2^M-1}$. The $N$ controlled agents $\mathcal{C}$ is a subset of a team of agents $\mathcal{M}$, and the $M-N$ uncontrolled agents $\mathcal{U} := \mathcal{M} \backslash \mathcal{C}$ is necessary for evaluating the controlled agents' policies (see Eq. 1). The ignorance of uncontrolled agents will reduce the $\mathcal{G}(s)$ to a subspace $W := \mathbb{R}^{2^N-1}$, and the external teamwork will be totally neglected. This introduces bias, as interpreted by the inter-vector angle before and after ignoring uncontrolled agents, which is explained in Example 3.4.*

**Example 3.4.** *As shown in Figure 1, suppose we have a team of agents $\mathcal{M} = \{1, 2\}$ for a state $s$, where $\mathcal{C} = \{1\}$ and $\mathcal{U} = \{2\}$. $\theta$ is the angle between two vectors representing two situations, such that $w_s \in W = span\{v^1_{\{1\},s}\}$ and $u_s \in \mathcal{G}(s) = span\{v^1_{\{1\},s}, v^1_{\{2\},s}, v^1_{\{1,2\},s}\}$. It is obvious that $w_s(\mathcal{M})$ as a projection of $u_s(\mathcal{M})$ loses information about the external teamwork.*

## 4 FROM SHAPLEY AXIOMS TO LEARNING FOR NAHT

In the following subsections, we show how to incorporate Markovian dynamics and how to extend the Shapley value's axioms to the state-specific cooperative game space for the NAHT. Together, these components establish an **axiomatic framework** in which individual value functions are shaped by structural constraints derived from the axioms. This framework provides a principled design space: enforcing all axioms yields Shapley values, while relaxing some leads to alternatives such as the Banzhaf index. Moreover, it shows a close connection between the Linearity axiom and TTD($\lambda$).

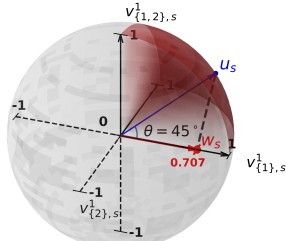

Figure 1: Example of state-specific cooperative games. The red and blue vectors are two games represented by different basis game sets, denoted by $w_s$ and $u_s$. Their difference is measured by the $\theta$. The shading area in red indicates the range of generated cooperative games with that $k_C \geq 0$ and $\sum (k_C)^2 = 1$.

### 4.1 A PRELIMINARY RESULT

By Definition 2.1, we have the expression $v^z_{C,s} = z v^1_{C,s}$, for any $z \in \mathbb{R}$, such that the condition that if $C \subseteq D$, $v^z_{C,s}(D) = z$, otherwise, $v^z_{C,s}(D) = 0$, still holds.

**Lemma 4.1.** *Given a fixed state $s \in \mathcal{S}$, each state-specific cooperative game's value $v_s \in \mathcal{G}(s)$ can be uniquely represented by: $v_s = \sum_{\emptyset \neq C \subseteq \mathcal{M}} k'_C \cdot v^z_{C,s}$, where $k'_C = \frac{k_C}{z}$ and $z \neq 0$.*

By Lemma 4.1, a cooperative game can be equivalently represented by the form: $v_s = \sum_{\emptyset \neq C \subseteq \mathcal{M}} k'_C \cdot v^{v_s(\mathcal{M})}_{C,s}$ with setting $z = v_s(\mathcal{M})$ and $k'_C = \frac{k_C}{v_s(\mathcal{M})}$, and $v_s(\mathcal{M}) \neq 0$.[2] Herein, the set of $v^{v_s(\mathcal{M})}_{C,s}$ become a set of new basis games and $k'_C$ are their corresponding weightings.

**Proposition 4.2.** *For the class of superadditive games formed by a set of basis games $\{v^z_{C,s} | \emptyset \neq C \subseteq \mathcal{M}\}$, it holds that $k_C \geq 0$, for all $\emptyset \neq C \subseteq \mathcal{M}$.*

Following discussion in Section 2.2, we aim to restrict the state-specific cooperative game class to the superadditive games. By Proposition 4.2, we can conclude that $k'_C \geq 0$ is a sufficient condition for $v_s$ belonging to the superadditive game class. **In other words, $k'_C \geq 0$ is a key condition for reaching cooperation within a team of agents.**

Introducing a **linear multidimensional operator** $\phi : \mathcal{G}(s) \to \mathbb{R}^M$ on a state-specific cooperative game $v_s$ defined by Lemma 4.1, we derive the following formula such that:

$$\phi(v_s) = \phi \left( \sum_{\emptyset \neq C \subseteq \mathcal{M}} k'_C \cdot v^{v_s(\mathcal{M})}_{C,s} \right), \tag{2}$$

---

[2]If $v_s(\mathcal{M}) = 0$, it implies that a team of agents unable to cooperate, contradicting the purpose of the NAHT.

where each $v_{C,s}^{v_s(\mathcal{M})}$ becomes a new basis game and $k_C'$ is its corresponding new weighting. By the Linearity axiom of $\phi$, we can transform Eq. 2 as follows:

$$\phi(v_s) = \sum_{\emptyset \neq C \subseteq \mathcal{M}} k_C' \cdot \phi\left(v_{C,s}^{v_s(\mathcal{M})}\right). \qquad (3)$$

## 4.2 Incorporating Markovian Dynamics into Game-Structured NAHT

For brevity, we rewrite $v_{C,s}^{v_s(\mathcal{M})}(\mathcal{M})$ as $V(C,s)$ for all $C \subseteq \mathcal{M}$. Then, we get the following fact:

**Fact 4.3.** $V(C_1, s) = V(C_2, s) = \cdots = V(\mathcal{M}, s) = v_s(\mathcal{M})$ *for all* $C_i \in \mathcal{P}_+(\mathcal{M})$, *as per Definition 2.1 and* $\mathcal{M} \supseteq C_i$, *where* $\mathcal{P}_+(\mathcal{M})$ *indicates the powerset of* $\mathcal{M}$ *excluding the empty set.*

Continued from the game-structured NAHT defined in Section 3, we now introduce Markovian dynamics to establish connection between state-specific cooperative game values of consecutive states, adapting to the environmental condition in the Dec-POMDP (Oliehoek et al., 2016). The only postulate is that each state-specific cooperative game value can be represented in the form of rewards, such that $v_{s_t}(\mathcal{M}) = \mathbb{E}_\pi[\sum_{\tau=0}^{T-t-1} \gamma^\tau R_{t+\tau} \mid s_t] \in \mathcal{G}(s)$, following the notation in Section 2.1. By the convention of RL (Sutton & Barto, 2018)[Chap. 7], it is feasible to use the **n-step return** to express the $v_{s_t}(\mathcal{M})$ under **a fixed horizon** $\bar{n}$, such that:

$$v_{s_t}(\mathcal{M}) = \mathbb{E}_\pi\left[\sum_{\tau=0}^{\bar{n}-1} \gamma^\tau R_{t+\tau} + \gamma^{\bar{n}} v_{s_{t+\bar{n}}}(\mathcal{M}) \;\middle|\; s_t\right] := \mathbb{E}_\pi\left[G_{t:t+\bar{n}} \mid s_t\right]. \qquad (4)$$

**Definition 4.4** (Size–Lexicographic Coalition Order and Horizon Assignment)**.** *Set* $m := 2^M - 1$. *Fix a size-lex enumeration* $(C_1, \ldots, C_m)$ *satisfying: (1)* $|C_i| \leq |C_{i+1}|$ *for all* $i$ *(nondecreasing by size); (2) within each size, ties are broken by a fixed total order (e.g., lexicographic over agent indices). Define horizons by* $n_1 := 1, n_i := n_{i-1} + 1 \; (i \geq 2)$. *Finally, set the basis game values as:*

$$V(C_i, s) := \mathbb{E}_\pi\big[G_{t:t+n_i} \mid s_t = s\big], \qquad (i = 1, \ldots, m).$$

We justify Definition 4.4 in Appendix C by illustrating the alignment between the variance ordering of $V(C_i, s)$ and that of $E_\pi\big[G_{t:t+n_i} \mid s_t = s\big]$.

**Proposition 4.5.** *Under Definition 4.4, for every* $s \in \mathcal{S}$,

$$\big(V(C_1, s), \ldots, V(C_m, s)\big) = \big(\mathbb{E}_\pi[G_{t:t+n_1} \mid s], \ldots, \mathbb{E}_\pi[G_{t:t+n_m} \mid s]\big),$$

*and, by Fact 4.3, each coordinate is cardinally equal to* $V(\mathcal{M}, s)$.

*Proof.* Immediate from the definition of $V(C_i, s)$; cardinal equality follows from Fact 4.3. $\qquad \square$

By Fact 4.3 and Proposition 4.5, we obtain the following formula that aligns basis game values of various coalitions to their n-step returns with corresponding horizons $1 \leq n \leq m$:

$$V(C_n, s_t) = \mathbb{E}_\pi\left[\sum_{\tau=0}^{n-1} \gamma^\tau R_{t+\tau} + \gamma^n V(\mathcal{M}, s_{t+n}) \;\middle|\; s_t\right], \forall \emptyset \neq C_n \subseteq C_{n+1}, C_m = \mathcal{M}. \qquad (5)$$

## 4.3 Linearity Axiom for Game-Structured NAHT

Recall that $v_{C_n, s_t}^{v_{s_t}(\mathcal{M})}(\mathcal{M})$ is denoted by $V(C_n, s_t)$, and $v_{s_t}(\mathcal{M})$ is denoted by $V(\mathcal{M}, s_t)$. Substituting Eq. 5 into the Linearity axiom in Eq. 3, we obtain the following formula:

$$\phi(V(\mathcal{M}, s_t)) = \sum_{n=1}^{m} k_{C_n}' \cdot \mathbb{E}_\pi\left[\sum_{\tau=0}^{n-1} \gamma^\tau \phi(R_{t+\tau}) + \gamma^n \phi(V(\mathcal{M}, s_{t+n})) \;\middle|\; s_t\right]. \qquad (6)$$

By linearity of the expectation operator, we obtain that:

$$\phi(V(\mathcal{M}, s_t)) = \mathbb{E}_\pi\left[\sum_{n=1}^{m} k_{C_n}' \cdot \sum_{\tau=0}^{n-1} \gamma^\tau \phi(R_{t+\tau}) + \gamma^n \phi(V(\mathcal{M}, s_{t+n})) \;\middle|\; s_t\right], \qquad (7)$$

where the term within the $\mathbb{E}_\pi[\cdot]$ takes the form of **truncated $\lambda$-return** (Sutton & Barto, 2018)[Chap. 12], if all $k'_{C_n}$ are seen as weightings for n-step return components.

In principle, we aim to learn $\phi_i(V(\mathcal{M}, s_t))$ directly for each controlled agent $i \in \mathcal{C}$. **For brevity, we represent each $\phi_i(V(\mathcal{M}, s_t))$ as $V_i(s_t)$, and $\phi_i(R_t)$ as $R_{t,i}$, respectively.** By sampling trajectories using a joint policy $\pi$, we derive the following TD($\lambda$) error $\delta_{t,i}$ (Sutton & Barto, 2018)[Chap. 12] for each agent $i$ such that:

$$\delta_{t,i} := \sum_{n=1}^{m} k'_{C_n} \cdot \sum_{\tau=0}^{n-1} (\gamma^\tau R_{t+\tau,i} + \gamma^n V_i(s_{t+n})) - V_i(s_t). \tag{8}$$

## 4.4 Efficiency Axiom for Game-Structured NAHT

**Proposition 4.6** (**Representation of Transformed Rewards**). *Given the condition $\sum_{i=1}^{M} \phi_i(R_t) = R_t$, the payoff allocation defined on rewards $R_t$, can be expressed as:*

$$\phi_i(R_t) := R_t - \sum_{j \neq i} (\phi_j(V(\mathcal{M}, s_t)) - \gamma \phi_j(V(\mathcal{M}, s_{t+1}))). \tag{9}$$

Note that $\phi(R_{t+\tau})$ in Eq. 7 has not yet been defined. For obeying the state-specific cooperative game space $\mathcal{G}(s)$ for the multidimensional operator $\phi(\cdot)$, it is necessary to represent $\phi(R_{t+\tau})$ in the form of $\phi(V(\mathcal{M}, s))$. To achieve this, we now introduce the Efficiency axiom to define $\phi(R_t)$. To satisfy the Efficiency axiom such that $\sum_{i=1}^{M} \phi_i(V(\mathcal{M}, s_t)) = V(\mathcal{M}, s_t)$, it is reasonable to presume its sufficient condition holds: $\sum_{i=1}^{M} \phi_i(R_t) = R_t$. Literally, each agent's value expansion can be expressed independently with its own $\phi_i(R_t)$, which will be detailed in the next subsection. Consequently, the resulting expression of $\phi_i(R_t)$ is presented in Proposition 4.6.

Substituting Eq. 9 into Eq. 8 with replacing $\phi_i(R_t)$ and $\phi_i(V(\mathcal{M}, s_t))$ by $R_{t,i}$ and $V_i(s_t)$, the $R_{t,i}$ in Eq. 8 can be rewritten as follows:

$$R_{t,i} = R_t - \sum_{j \neq i} (V_j(s_t) - \gamma V_j(s_{t+1})). \tag{10}$$

**Theorem 4.7** (**Principle of Decomposing Value Functions** (Oliehoek et al., 2016)). *Given an additively factored reward function, for any timestep $t$ there is a factorization of the transition function, such that the value of a finite-horizon factored Dec-POMDP is decomposable across agents.*

The condition $R_t = \sum_{i=1}^{M} R_{t,i}$ for deriving the Efficiency axiom is also referred to as additively factored immediate reward function. Theorem 4.7 indicates that the Efficiency axiom always exists in the Dec-POMDP, given a factorization scheme of the transition function in addition to an additively factored immediate reward function. Recall that the NAHT is considered in the Dec-POMDP (see Section 2.1). To guarantee the feasibility of the Efficiency axiom, another constraint is added to implicitly facilitate searching for a factorization of the transition function such that:

$$\sum_{i=1}^{M} V_i(s_t) = V(\mathcal{M}, s_t). \tag{11}$$

## 4.5 Symmetry Axiom for Game-Structured NAHT

**Theorem 4.8.** *The payoff allocation operator satisfying permutation-equivariance is a sufficient condition for the Symmetry axiom.*

The Symmetry axiom means that two agents who contribute equally to every possible coalition (excluding themselves) should receive equal payoff allocations (Chalkiadakis et al., 2011)[Chap. 2]. In the literature from Dubey (1975), it was described as the permutation-equivariance of an multidimensional operator $\phi$ defined on the cooperative game value function, since **permutation-equivariance is a sufficient condition for the Symmetry axiom** (see Theorem 4.8). As a result, **we will focus on how to construct such an operator fulfilling permutation-equivariance**. Mathematically, we can express this as: $\phi_{\sigma(i)}(v_s) = \phi_i(\sigma(v_s))$, where $\sigma$ means relabelling agent IDs to

another ordering, e.g., from $(1, 2)$ to $(2, 1)$. Recall that in the settings of NAHT, the environment is defined as a Dec-POMDP, where the transition function is defined as $P_T(s_{t+1}|s_t, a_t)$ with $a_t$ being a joint action set, implying that the environment is invariant to exchanging agent identities and the cooperative game values are identical under exchanging agent identities. For this reason, $v_s = \sigma(v_s)$ naturally holds. We now replace $\phi$ with $V_i$ to keep consistency of notations as the above two axioms. The rest of requirement to make the permutation-equivariance hold is that the operator's output is changed consistently with permuting agent IDs, e.g., $(1, 2) \mapsto (2, 1) \Rightarrow (V_1, V_2) \mapsto (V_2, V_1)$. This implies that the operator's output of each agent ID should be invariant to permutation of its input, e.g., $V_i(\sigma(inp)) = V_i(inp)$.

**Remark 4.9.** *The structure of the operator for payoff allocations for fulfilling the Symmetry axiom is as follows: (1) The operator's output is changed consistently with permuting agent IDs. (2) The operator's output of each agent ID should be invariant to permutation of its input.*

### 4.6 FROM SHAPLEY AXIOMS TO LEARNING ALGORITHMS

**Theorem 4.10.** *Shapley Machine is an algorithm enforcing $V_i$ to fulfil Efficiency, Symmetry and Linearity, so the $V_i$ is the Shapley value for dynamic scenarios.*

The algorithm that enables individual values $V_i$ to fulfil the Efficiency, Linearity and Symmetry axioms through learning is named **Shapley Machine**: (1) Shape individual rewards following Eq. 10 and set Eq. 11 as a regularization term, to fulfil the Efficiency axiom. (2) Implement the TD error following Eq. 8, to fulfil the Linearity axiom. The $k'_C > 0$ is implemented as geometric distribution following the convention of TD($\lambda$) (Sutton & Barto, 2018)[Chap. 12]. (3) Structure the policy and value networks to fulfil the Symmetry axiom. Since the Linearity axiom is a sufficient condition for the Additivity axiom as mentioned in Section 2.2, the individual values $V_i$ satisfying all those axioms in implementation realizes Shapley values for dynamic scenarios, as highlighted in Theorem 4.10. In implementation, we use two base algorithms satisfying the conditions of partial observations: POAM (Wang et al., 2025) and IPPO (De Witt et al., 2020) to realize Shapley Machine, referred to as: **SM-POAM** and **SM-IPPO**, respectively. The details of implementation are left to Appendix E.1.

**Remark 4.11.** *Banzhaf Machine is an algorithm generating $V_i$, fulfilling Symmetry and Linearity.*

Following the same principle of designing algorithms via axiomatic characterization, we also propose a variant that enforces the Linearity and Symmetry axioms while omitting Efficiency. Since the resulting $V_i$ aligns with the Banzhaf index (Banzhaf III, 1964)[3], we refer to this algorithm as the **Banzhaf Machine**. Analogous to the Shapley Machine implementations above, its instantiations on POAM and IPPO are denoted as **BM-POAM** and **BM-IPPO**, respectively.

## 5 EXPERIMENTS

In experiments, we evaluate the proposed SM-POAM, SM-IPPO, BM-POAM and BM-IPPO on **the modified MPE-PP and SMAC tasks tailored to the NAHT settings** (Wang et al., 2025), against the counterpart baseline algorithms POAM and IPPO. The uncontrolled agents are taken from five pre-trained policies, which were trained in fully controllable MARL settings using the algorithms IPPO, MAPPO, VDN, QMIX, and IQL. To avoid confusion between the standard IPPO algorithm (trained under fully controllable MARL settings) and the version trained within the NAHT process (where only a subset of agents is controllable), we refer to the latter as IPPO-NAHT. Also, we validate the game-structured NAHT, a core concept in our theory, by setting the number of n-step return components considered in TTD($\lambda$) using the number of basis games $m$. Since the number of basis games is far more than the preset episode length (511 vs. 150) in 10v11, we approximate the number of basis games to the episode length as a trade-off. About the 8v9 scenario, the number of basis games needed in theory is close to the episode length (127 vs. 120), so the number of basis games can be naturally approximated by the episode length. As a result, **for 8v9 and 10v11 Banzhaf Machine is equivalent to the baseline counterparts in implementation**. The details of baselines, experimental settings and evaluation metrics are provided in Appendix E. All results are obtained by first computing each metric with 128 episodes and then averaging these per-seed metrics across 5 random seeds, with 95% confidence intervals reported.

---

[3]Banzhaf index is a payoff allocation that satisfies Symmetry and Linearity (Additivity), but not vice versa.

**Research Questions.** We focus on the following research questions in experimentation: (1) Is the number of coalitions $m$ used to construct TTD($\lambda$) as defined in Proposition 4.5 a critical factor for learning performance? Answering this question provides empirical support for the rationality of our theoretical framework. (2) Are all the axioms characterizing the Shapley value equally appropriate across scenarios? This question highlights the flexibility of our axiomatic framework: rather than relying directly on the explicit Shapley formula as in prior work, we treat its axioms as design elements that can be selectively enforced or relaxed depending on the task. (3) Which axiom is key to generalization to unseen conventions or unseen agent types? The answer can emphasize the practicability of our axiomatic perspective.

## 5.1 MAIN RESULTS AND ANALYSIS

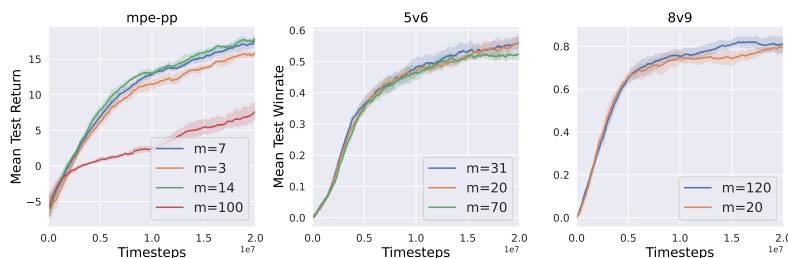

Figure 2: Experiments verifying that the number of n-step return components in TTD($\lambda$) can be determined by the number of non-empty basis games. In MPE-PP the number of basis games is $m = 7$, in 5v6 it is $m = 31$, and in 8v9 it is $m = 120$.

**Answer to Question 1.** We conduct experiments with varying numbers of basis games $m$ using SM-POAM as the candidate algorithm. As shown in Figure 2, learning performance in MPE-PP and 8v9 is sensitive to the choice of $m$, while in 5v6 it is comparatively robust. Overall, these results indicate that **setting the number of n-step return components as the number of non-empty basis games can nearly maximize learning performance**, thereby providing an empirical evidence to support the mathematical rationale behind Proposition 4.5. Although alternative theories or frameworks may exist for different choices of $m$—as seen in the good performance of MPE-PP and 5v6—**this does not undermine the validity of Proposition 4.5**.

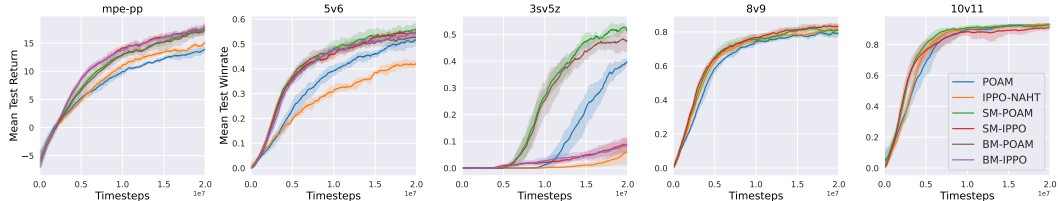

Figure 3: Evaluation curves during training in NAHT.

**Answer to Question 2.** We evaluate all baselines alongside our proposed algorithms. For the large-scale tasks 8v9 and 10v11, the base algorithms IPPO-NAHT and POAM are effectively equivalent to BM-IPPO and BM-POAM under the approximation of $m$, due to limited episode lengths, and are therefore omitted as separate implementations. Figure 3 shows that, for either IPPO-NAHT or POAM, at least one of their Shapley Machine or Banzhaf Machine variants achieves the best performance. Recall that Banzhaf Machine corresponds to dropping the Efficiency axiom. In MPE-PP and 3sv5z with IPPO-NAHT as the base, Banzhaf Machine performs on par with Shapley Machine. In most other scenarios (except 10v11 with IPPO-NAHT), Shapley Machine outperforms Banzhaf Machine, although sometimes by a narrow margin. Moreover, comparisons between the base algorithms and their Banzhaf counterparts in MPE-PP, 5v6, and 3sv5z underscore the importance of the Linearity axiom. Overall, the results suggest that **all three Shapley axioms generally enhance learning performance (in possibly unseen team compositions), but the Efficiency axiom can, in some cases, reduce it**. This confirms the value of our proposed axiomatic framework, which flexibly accommodates both Shapley- and Banzhaf-style formulations.

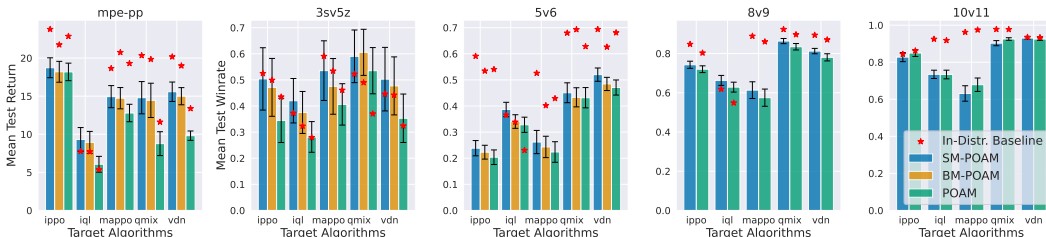

(a) Evaluation on unseen conventions (identical algorithms with different random seeds). The star denotes the in-convention baseline, and the histograms show performance on unseen conventions with error bars.

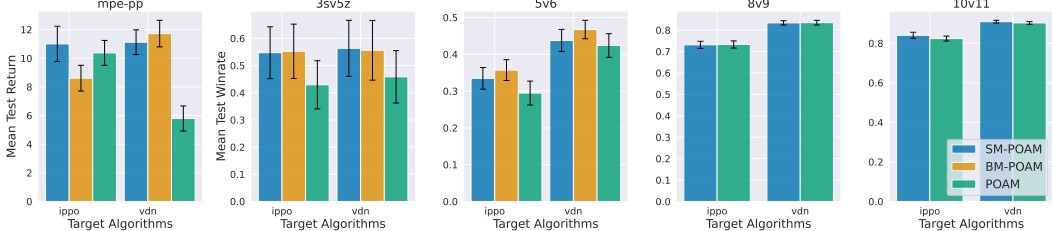

(b) Evaluation on unseen agent types. Train set: MAPPO, QMIX, IQL; test set: IPPO, VDN.

Figure 4: Out-of-distribution (OOD) evaluation after training, assessing unseen conventions and unseen agent types during training, between learned agents (POAM, SM-POAM and BM-POAM) and uncontrolled agent types (IPPO, VDN, IQL, QMIX and MAPPO). The performance is evaluated by averaging all pairings of $N$ controlled and $M - N$ uncontrolled agents and their corresponding random seeds, called M–N score (Wang et al., 2025).

**Answer to Question 3.** We evaluate all baselines alongside our proposed algorithms after training, to test their capability of generalization to unseen conventions and unseen agent types, respectively. The evaluation is conducted by the best model saved during training for each random seed of running experiments. It can be seen from Figure 4a that SM-POAM outperforms both BM-POAM and POAM on all scenarios, except for the 10v11 due to the weak learning performance. This implies that **in general all Shapley axioms may be instrumental in tackling unseen conventions**. As seen from Figure 4b, the performance of BM-POAM is comparable to and even better than SM-POAM in most cases. This implies that **the Efficiency axiom could weaken the capability of generalization to unseen agent types**. This phenomenon recurs, and becomes even more pronounced, in the evaluation of IPPO-NAHT related algorithms, as shown in Figure 10b in Appendix F.3.

## 5.2 Number and Weightings of Basis Game Values

The weightings of basis game values ($k'_C$ in Eq. 8) are set up by the probability values of a geometric distribution governed by $\lambda$ in this paper, respecting the convention of RL. The $\lambda$ will influence the shape of the geometric distribution. When the $\lambda$ grows larger, the geometric distribution tends to be a longer tail with the probability decays slowly as the number of basis games $m$ increases, and vice versa. In other words, **controlling $m$ can be implemented by simply changing the weightings of basis game values, or equivalently, the $\lambda$, under the assumption of the geometric distribution**. To verify this result holding in the NAHT, we conduct a case study, as shown in Figure 5. It can be observed that the number of effective basis games (the probability values over which are non-zero) for $m = 31$ and $m = 70$ is the same under $\lambda = 0.85$, resulting in nearly identical learning performance. **This aligns with the conventional view of $\lambda$'s impact on TTD($\lambda$) in RL, primarily as a mechanism for controlling the tail length of the return (Sutton & Barto, 2018)[Chap. 12.3].**

Moreover, we demonstrate that even with the same number of basis games, the values of their corresponding weightings significantly influence the learning performance. As shown in Figure 6, given the same number of active basis games as $m = 7$, the learning processes with $\lambda = 0.5$ and $\lambda = 0.85$ result in diverse performance. **This provides an alternative insight into how $\lambda$ influences TTD($\lambda$), diverging from the traditional RL perspective.**

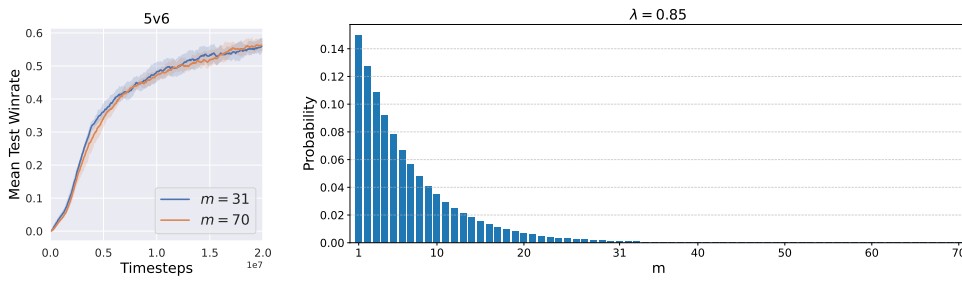

(a) SM-POAM: $m = 31$ vs. $m = 70$ for $\lambda = 0.85$.

(b) Demonstration of the shape of weightings of basis game values with $\lambda = 0.85$.

Figure 5: Controlling the number of effective basis games can be implemented by changing the parameters to control the shape of weightings of basis game values. Note that the probability values for $m > 31$ are nearly zero.

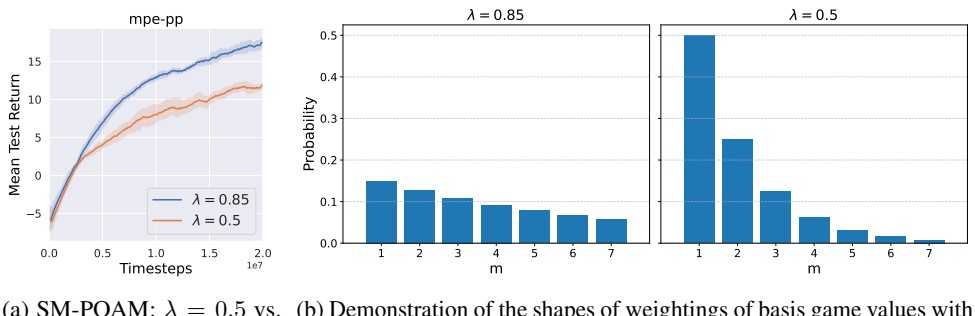

(a) SM-POAM: $\lambda = 0.5$ vs. $\lambda = 0.85$ for $m = 7$.

(b) Demonstration of the shapes of weightings of basis game values with $m = 7$, for $\lambda = 0.85$ and $\lambda = 0.5$, respectively.

Figure 6: Influence of different values of basis games given the same number of effective basis games. Note that the shapes of geometric distributions vary significantly with different values of $\lambda$, under the same number of effective $m$.

**Final Remark.** Based on the above findings, the main results are obtained by the algorithms with $\lambda$ that enables weightings to contribute to the effective number of basis games as $m$. In the future work, it is valuable to investigate how to generate $\lambda$ or broader classes of weighting functions, so that it can control the number of effective basis games (n-return components) for the NAHT.

## 6 CONCLUSION

**Summary.** This paper addresses the NAHT through the lens of cooperative game theory. We propose an axiomatic framework that models individual value functions through the Shapley axioms (Additivity, Efficiency, Symmetry), leading to the Shapley Machine, which uniquely recovers Shapley values. Unlike prior MARL work based on the explicit Shapley formula, our approach reveals their connection to TTD($\lambda$) and establishes a design space: enforcing all axioms yields Shapley-based methods, while relaxing them gives alternatives such as the Banzhaf Machine.

**Limitation and Future Work.** Although our work primarily aims to introduce an axiomatic framework for NAHT, it has several limitations. First, the Efficiency axiom shows limited scalability, as it does not perform well in large-scale multi-agent scenarios. Second, the physical interpretation of Proposition 4.5 remains open, leaving its practical grounding incomplete. These limitations motivate further exploration. A key contribution of our framework is the introduction of state-specific cooperative game models, where the Linearity axiom provides a game-theoretic explanation for the weightings of $n$-step return components in TTD($\lambda$) (Cichosz, 1994). The preliminary case studies in Section 5.2 suggest a promising research direction: **exploring broader classes of weighting functions beyond the standard geometric distribution**. For instance, weightings $k_C$ satisfying $\sum (k_C)^2 = 1$ (as illustrated in Figure 1) could define alternative weighting schedules.

## REPRODUCIBILITY STATEMENT

This paper is mixed of theoretical and algorithm contributions. To let the readers understand our main contributions better, we present extra background knowledge in Appendix B. For all mathematical claims in the paper, their justifications and proofs can be found in Appendices C and D. The implementation details of our algorithms and baselines are shown in Appendix E.1. The metrics for evaluating results, experimental settings and hyperparameters used in experiments are shown in Appendices E.3, E.4 and E.5, respectively. The computational resources usage and the time for experiments are exposed in Appendix E.6. Finally, we have uploaded an anonymous version of codes to supplementary materials.

## THE USE OF LARGE LANGUAGE MODELS

We made limited use of large language models (LLMs) during the preparation of this paper. Specifically, LLMs were employed to assist in searching for related work, polishing the language of the manuscript, and providing clarifications on background knowledge with which the authors were less familiar. Importantly, LLMs did not contribute to the design of the research, the development of the methodology, or the generation of research ideas. All content suggested by LLMs was carefully reviewed and verified by the authors before inclusion in the final version of the paper.

As for polishing the language of the manuscript, every time the authors only input several sentences or a short paragraph (rather than the whole draft) written by themselves to LLMs. When LLMs returned the polished words, the authors reviewed the preciseness and authenticity of the contents generated.

About searching for related work and background knowledge, every time the authors input a query about the research area of interest and received a summary from LLMs. Then, the authors manually and carefully checked the materials informed by LLMs, ensuring that LLMs only played the role as an assistant or an intelligent search engine.

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

## A  RELATED WORK

### A.1  TD($\lambda$) IN REINFORCEMENT LEARNING AND MULTI-AGENT REINFORCEMENT LEARNING

Related work on temporal-difference learning with eligibility traces centers on the seminal TD($\lambda$) algorithm proposed by Sutton (1988), which unified one-step TD and Monte Carlo methods via a trace-decay parameter $\lambda$ and introduced the forward- and backward-view equivalence for efficient multi-step credit assignment. Subsequent theoretical analyses (Dayan, 1992; Dayan & Sejnowski, 1994; Tsitsiklis & Van Roy, 1996) established convergence guarantees and characterized the bias–variance trade-off inherent in $\lambda$-returns, while extensions by Watkins et al. (1989); Peng & Williams (1994) adapted eligibility traces for off-policy control, Q($\lambda$). Advances in function approximation, including true online TD($\lambda$) (Seijen & Sutton, 2014) and gradient-TD methods, further broadened applicability to large-scale and nonlinear settings, inspiring n-step return techniques such as generalized advantage estimation (GAE) (Schulman et al., 2015) for policy optimization. In multi-agent reinforcement learning, algorithms of PPO family such as MAPPO (Yu et al., 2022) and IPPO (De Witt et al., 2020), applied GAE to optimize policies and thus used TD($\lambda$) as the target values to train critics. This paper highlights the theoretical link between truncated TD($\lambda$), denoted TTD($\lambda$) (Cichosz, 1994), in NAHT (a generalization of MARL) and the axioms of the Shapley value. It further shows that the number of $n$-step return components in TTD($\lambda$) can be determined by the number of agents. This insight provides a theoretical foundation that could inform the design of a broader class of multi-agent reinforcement learning algorithms.

### A.2  THEORETICAL MODELS FOR AD HOC TEAMWORK

We now discuss theoretical models describing ad hoc teamwork (AHT). Brafman & Tennenholtz (1996) pioneered research into AHT by investigating repeated matrix games involving a single teammate. Subsequent studies expanded this framework to scenarios with multiple teammates, notably by Agmon & Stone (2012). Later, Agmon et al. (2014) further relaxed earlier assumptions by allowing teammates' policies to be selected from a known set. Stone et al. Stone & Kraus (2010) initially formalized AHT through collaborative multi-armed bandits, albeit under notable assumptions such as prior knowledge of teammates' policies and environmental conditions. Albrecht & Ramamoorthy (2013) advanced this field significantly by introducing the stochastic Bayesian game (SBG), the first comprehensive theoretical framework accommodating dynamic environments and unknown teammate behaviours in AHT. Building on SBG, Rahman et al. (2021) proposed the open stochastic Bayesian game (OSBG), addressing open ad hoc teamwork (OAHT). Zintgraf et al. (2021) modelled AHT through interactive Bayesian reinforcement learning (IBRL) within Markov games, specifically targeting non-stationary teammate policies within single episodes. Xie et al. (2021) introduced the hidden parameter Markov decision process (HiP-MDP) to handle situations where teammates' policies vary across episodes but remain stationary during individual episodes. Most recently, Wang et al. (2024) extended OSBG by incorporating principles from cooperative game theory, introducing the open stochastic Bayesian coalitional affinity game (OSBG-CAG), which theoretically justifies a graph-based representation for joint Q-value functions and includes rigorous convergence proofs for Q-learning algorithms in open team settings. This paper adopts the perspective of cooperative game theory, introducing an axiomatic framework based on state-specific cooperative games to address the NAHT problem. Within this framework, the Shapley value's axioms are extended and enforced, so that Shapley-based algorithms emerge implicitly by satisfying these axioms during learning. More importantly, this axiomatic characterization also motivates a new algorithm, the Banzhaf Machine, which preserves the advantages of the Shapley value while alleviating some of its limitations.

### A.3  SHAPLEY VALUE IN MULTI-AGENT REINFORCEMENT LEARNING

Related work mainly focused on developing the theory of Shapley value in MARL, and incorporated Shapley value (Shapley, 1953) into MARL algorithms. Early studies (Wang et al., 2020) incorporated Shapley value into credit assignment scheme in a principled way and proposed an algorithm named SQDDPG, underpinned by the equivalence between cooperative-game theoretical models in dynamic scenarios and the shared reward Markov games. Wang et al. (2022) further improved the theory by proving the existence of Shapley value, and proposed an algorithm named SHAQ, an algorithm with promising convergence to find an optimal joint policy. It also shed light on the relationship

between Shapley value and the relevant value decomposition and credit assignment approaches for MARL. Li et al. (2021) improved the stability of SQDDPG and proposed an algorithm named Shapley Counterfactual Credits. Han et al. (2021) incorporated Shapley value into the model-based PPO as each agent's advantage value, and estimated coalition values by trained transition and reward models. Chai et al. (2024) continued the idea from Han et al. (2021), by replacing the the transition and reward models with more powerful world models. Xue et al. (2022) incorporated Shapley value into multi-agent communication, as a criterion for forming communication between agents. This paper primarily focuses on learning the Shapley value to address the NAHT, a generalized paradigm to MARL. More importantly, the Shapley values are learned by complying with their axioms, rather than by constructing their explicit formulas as implemented in the previous work. This may help overcome the notorious learning instability issues caused by learning substantial "coalitional values" in the existing algorithms based on Shapley values. Furthermore, the insight of axiomatic characterization can facilitate the flexibility of Shapley value approaches in a wide variety of tasks. Specifically, one can safely drop any part of a learning algorithm, if it is justified that the targeting task does not align with the corresponding axiom.

## B    EXTRA BACKGROUND

### B.1    $\lambda$-RETURN AND TD($\lambda$)

We now introduce an extension of return named $\lambda$-return. Mathematically, a $\lambda$-return $G_t^\lambda$ for infinite-horizon cases can be expressed as follows:

$$G_{t:t+n} = R_t + \gamma R_{t+1} + \cdots + \gamma^{n-1} R_{t+n-1} + \gamma^n V(s_{t+n}),$$
$$G_t^\lambda = (1 - \lambda) \sum_{n=1}^{\infty} \lambda^n G_{t:t+n}. \tag{12}$$

Similarly, the $\lambda$-return for finite-horizon cases can be expressed as follows:

$$G_t^\lambda = (1 - \lambda) \sum_{n=1}^{T-t} \lambda^n G_{t:t+n} + \lambda^{T-t} G_{t:T}. \tag{13}$$

Note that if $\lambda = 1$, updating according to the $\lambda$-return is a Monte Carlo algorithm. In contrast, if $\lambda = 0$, then the $\lambda$-return reduces to $G_{t:t+1}$, the one-step return. Monte Carlo algorithm is known as its high variance but low bias, while one-step return is known as its low variance but high bias. For this reason, the change of $\lambda$ can be seen as a tradeoff between bias and variance. The TD prediction using $\lambda$-return as the target value is referred to as TD($\lambda$) (Sutton, 1988).

**A variant of TD($\lambda$) by shaping $G_t^\lambda$ with a designated horizon $t < H \leq T$ is referred to as truncated TD($\lambda$), shortened as TTD($\lambda$)** (Cichosz, 1994). Mathematically, $G_t^\lambda$ with a designated horizon $H$ can be formulated as follows:

$$G_t^\lambda = (1 - \lambda) \sum_{n=1}^{H-t} \lambda^n G_{t:t+n} + \lambda^{H-t} G_{t:H}. \tag{14}$$

Note that in the convention of RL, the horizon $H$ is defined as the time difference from the starting point of an episode, denoted by $t = 0$.

**Remark B.1.** *In this paper, to simply the presentation of our work, we redefine the horizon as the time difference from each timestep $t$, denoted as $m$, such that $m = H - t$. With the new definition of horizon, we can rewrite the above formula of $G_t^\lambda$ as follows:*

$$G_t^\lambda = (1 - \lambda) \sum_{n=1}^{m} \lambda^n G_{t:t+n} + \lambda^m G_{t:t+m}. \tag{15}$$

### B.2    SUPERADDITIVE GAME AND COOPERATIVE MULTI-AGENT REINFORCEMENT LEARNING

The superadditive game is a subclass of characteristic games that satisfies an additional condition: $v(C \cup D) \geq v(C) + v(D)$, for any two distinct coalitions $C, D \subseteq \mathcal{M}$ with $C \cap D = \emptyset$. Previous

work in MARL (Wang et al., 2020) has shown the equivalence on the task objective function between state-based superadditive games with action space and team reward Markov games with coalition structures.[4] Intuitively, the Shapley value is a solution to the stability of a team formation, resembling cooperation. Since the NAHT is a generalization of MARL, this conclusion still valid in principle. It has been already proved that the Shapley value exists in superadditive games (Shapley, 1953) and each game instance in superadditive games can be uniquely represented as $\sum k_C v_C^1$, where $k_C \geq 0$ (Dubey, 1975). In the cooperative game theory, any game can be transformed to a superadditive game if its value $v$ is transformed by a *superadditive cover* $v^*$ such that $v^*(C) = \max_{CS(C)} \sum_{D \in CS(C)} v(D)$ (Greco et al., 2011). As a result, **it will not lose generality if we consider the cooperative game space as superadditive games in this paper**.

### B.3 Baseline Algorithms: POAM and IPPO

**POAM.** In general, POAM is an algorithm built upon IPPO (De Witt et al., 2020). The primary difference between POAM and IPPO is as follows: **For each agent's policy and value, in addition to its observation used in IPPO, POAM takes as input an embedding vector to predict other agents' potential behaviours**. More specifically, this embedding vector is trained by an encoder-decoder structure, wherein the encoder is modeled as RNNs and decoder is modeled as MLPs. For brevity, $-i$ denotes the set of all agents excluding agent $i$. Let $h_{t,i} = \{o_{k,i}, a_{k-1,i}\}_{k=1}^{t}$ denote agent $i$'s history of observations and actions up to timestep $t$ and $e_{t,i} \in \mathbb{R}^n$ denote the resulting embedding vector of $n$ dimensions. The encoder parameterized by $\theta^e$ is defined as $f_{\theta^e}^{enc} : \mathcal{H}_i \to \mathbb{R}^n$. The embedding vector is decoded by two decoder networks: the observation decoder $f_{\theta^o}^{dec} : \mathbb{R}^n \to \mathcal{O}_{-i}$, and the action decoder $f_{\theta^a}^{dec} : \mathbb{R}^n \to \Delta(\mathcal{A}_{-i})$. The decider networks are respectively trained to predict the observations and actions of all other agents on the team at timestep $t$, $o_{t,-i}$ and $a_{t,-i}$, to encourage $e_{t,i}$ to contain information about collective behaviors corresponding to $h_{t,i}$. While the observation decoder directly predicts the observed $-i$'s observations, the action decoder predicts the parameters of a probability distribution over the $-i$'s actions $\pi_{-i}(a_{t,-i}; f_{\theta^a}^{dec}(f_{\theta^e}^{enc}(h_{t,i})))$. As we consider continuous observations and discrete actions, the loss function with using categorical distribution to model $\pi_{-i}$ is as follows:

$$L_{\theta^e,\theta^o,\theta^a}(h_{t,i}, o_{t,-i}, a_{t,-i}) = ||f_{\theta^o}^{dec}(f_{\theta^e}^{enc}(h_{t,i})) - o_{t,-i}||^2 - \log \pi_{-i}(a_{t,-i}; f_{\theta^a}^{dec}(f_{\theta^e}^{enc}(h_{t,i}))). \quad (16)$$

**For brevity, we define $g(h) := (h, f_{\theta^e}^{enc}(h))$ in the following description.**

POAM employed generalized advantage estimation (GAE) (Schulman et al., 2015) to form the policy gradient, and the value function used to form GAE is trained by the following loss function:

$$L_{\theta^c}(h_{t,i}) = \frac{1}{2}\left(V_i^{\theta^c}(g(h_{t,i})) - \hat{V}_{t,i}\right)^2, \quad (17)$$

where $\hat{V}_{t,i}$ is the finite-horizon TD($\lambda$) return, for which the horizon is up to the episode length $T$. Each agent $i$'s GAE estimation is as follows:

$$A_{t,i} = \sum_{l=0}^{T}(\gamma\lambda)^l \delta_{t+l,i}, \quad (18)$$

where $\delta_{t,i} = R_t + \gamma V_i(g(h_{t,i})) - V_i^{\theta^c}(g(h_{t,i}))$ and $V_i(g(h_{t,i}))$ denotes the non-parameterized individual value. Based on the GAE defined above, the policy optimization loss is defined as:

$$L_\theta(h_{t,i}, a_{t,i}) = \min\left\{\frac{\pi_\theta(a_{t,i}|g(h_{t,i}))}{\pi_{\theta_{old}}(a_{t,i}|g(h_{t,i}))}A_{t,i}, \text{clip}\left(\frac{\pi_\theta(a_{t,i}|g(h_{t,i}))}{\pi_{\theta_{old}}(a_{t,i}|g(h_{t,i}))}, 1-\epsilon, 1+\epsilon\right)A_{t,i}\right\}. \quad (19)$$

**IPPO.** By removing the loss for the encoder-decoder structure as shown in Eq. 16 and replacing the input $g(h_{t,i})$ by $h_{t,i}$ in the above set of equations, we get the loss functions of IPPO.

---

[4]In Wang et al. (2020), state-based convex games with action space were proved to be equivalent to team reward Markov games. However, setting $C \cap D = \emptyset$ can relax a convex game to a superadditive game.

## C  JUSTIFICATION AND DISCUSSION OF SIZE–LEXICOGRAPHIC COALITION ORDER AND HORIZON ASSIGNMENT

In this section, we will justify the insight behind the size–lexicographic coalition order and horizon assignment, as shown in Definition 4.4. In the beginning, we define a discrete-time stochastic process of TD error terms $(Z_n)_{n \in \mathbb{N}_0}$, where $v_\pi(s_t)$ denotes a generic state value function:

$$Z_{n+1} := R_{t+n} + \gamma v_\pi(s_{t+n+1}) - v_\pi(s_{t+n}), \quad \text{so} \quad G_{t:t+n+1} = G_{t:t+n} + \gamma^n Z_{n+1}.$$

We first show a generic result related to the one-step increment term in Lemma C.1. Then, we make the coalition-variance monotonicity assumption in Assumption C.2, which can be justified by Remark C.3, **with evidence from the basis game values of the game-structured NAHT**, as shown in Definition 3.2. By Lemma C.1 and Assumption C.2, we get that for the coalition enumeration $\{i\}$ there exist nondecreasing orderings of variances for both the sequence of basis game values of coalitions nondecreasing by sizes and the sequence of n-step returns with the horizon schedule $n_i = n_{i-1} + 1$. This motivates that we can define the following term under this **monotone-variance regularity**:

$$V(C_i, s) := \mathbb{E}_\pi\big[G_{t:t+n_i} \mid s_t = s\big], \qquad (i = 1, \ldots, m).$$

The justification of Definition 4.4 completes.

Remark C.5 highlights that alternative horizon schedules (e.g., $n_i = n_{i-1} + k, \quad n_1 = 1, k > 0$) can be incorporated, offering additional design possibilities within our axiomatic framework.

**Lemma C.1.** *If $(Z_n)$ has nonnegative autocovariances given $s_t = s$, then $n \mapsto \mathrm{Var}(G_{t:t+n} \mid s)$ is nondecreasing, with $\mathrm{Var}(G_{t:t+n+1} \mid s) = \mathrm{Var}(G_{t:t+n} \mid s)$ iff $\mathrm{Var}(Z_{n+1} \mid s) = 0$.*

*Proof.* Use $G_{t:t+n+1} = G_{t:t+n} + \gamma^n Z_{n+1}$ and expand the variance conditioning on $s_t = s$, we can get the following formula:

$$\mathrm{Var}(G_{t:t+n+1} \mid s) = \mathrm{Var}(G_{t:t+n} \mid s) + 2\gamma^n \mathrm{Cov}\big(G_{t:t+n}, Z_{n+1} \mid s\big) + \gamma^{2n} \mathrm{Var}(Z_{n+1} \mid s).$$

**We prove if $(Z_n)$ has nonnegative autocovariances given $s_t = s$, then $n \mapsto \mathrm{Var}(G_{t:t+n} \mid s)$ is nondecreasing.**

Using the sum of TD errors to represent the Monte Carlo error (Sutton & Barto, 2018)[Chap. 6], we have the following formula:

$$G_{t:t+n} - v_\pi(s_t) = \sum_{k=0}^{n-1} \gamma^k Z_{k+1}.$$

Given the linear property of covariance and a fact that the covariance of a constant is zero, we can get:

$$\mathrm{Cov}\big(G_{t:t+n}, Z_{n+1} \mid s\big) = \sum_{k=0}^{n-1} \gamma^k \mathrm{Cov}(Z_{k+1}, Z_{n+1} \mid s).$$

Therefore, **if all autocovariances $\mathrm{Cov}(Z_i, Z_j \mid s) \geq 0$**, then $\mathrm{Cov}\big(G_{t:t+n}, Z_{n+1} \mid s\big) \geq 0$.

Since $\mathrm{Var}(Z_{n+1} \mid s) \geq 0$ must hold, we can directly get:

$$\mathrm{Var}(G_{t:t+n+1} \mid s) \geq \mathrm{Var}(G_{t:t+n} \mid s).$$

That is, $n \mapsto \mathrm{Var}(G_{t:t+n} \mid s)$ is nondecreasing.

**We prove $\mathrm{Var}(G_{t:t+n+1} \mid s) = \mathrm{Var}(G_{t:t+n} \mid s)$ iff $\mathrm{Var}(Z_{n+1} \mid s) = 0$.**

It is a fact that $\mathrm{Var}(G_{t:t+n+1} \mid s) = \mathrm{Var}(G_{t:t+n} \mid s)$ iff

$$\mathrm{Var}(Z_{n+1} \mid s) = 0, \qquad \mathrm{Cov}\big(G_{t:t+n}, Z_{n+1} \mid s\big) = 0.$$

We aim to prove that $\mathrm{Var}(Z_{n+1} \mid s) = 0$ is sufficient for $\mathrm{Cov}\big(G_{t:t+n}, Z_{n+1} \mid s\big) = 0$.

If a nonnegative random variable (r.v.) $W \geq 0$ satisfies $\mathbb{E}[W \mid s] = 0$, then $W = 0$ almost surely (a.s.) given $s$ (Williams, 1991)[Chap. 9].

Apply this with $W = (Z_{n+1} - \mathbb{E}[Z_{n+1} \mid s])^2$. **If $\mathbf{Var(Z_{n+1} \mid s) = 0}$**, then

$$\mathbb{E}\left[(Z_{n+1} - \mathbb{E}[Z_{n+1} \mid s])^2 \mid s\right] = 0 \Rightarrow (Z_{n+1} - \mathbb{E}[Z_{n+1} \mid s])^2 = 0 \text{ a.s.} \Rightarrow Z_{n+1} = \mathbb{E}[Z_{n+1} \mid s] \text{ a.s.}$$

The covariance $\mathrm{Cov}(G_{t:t+n}, Z_{n+1} \mid s)$ can be expressed as the following formula:

$$\mathrm{Cov}(G_{t:t+n}, Z_{n+1} \mid s) = \mathbb{E}\left[\left(G_{t:t+n} - \mathbb{E}[G_{t:t+n} \mid s]\right)\left(Z_{n+1} - \mathbb{E}[Z_{n+1} \mid s]\right) \mid s\right].$$

Since $Z_{n+1} - \mathbb{E}[Z_{n+1} \mid s] = 0$ a.s., we get the following result:

$$\mathrm{Cov}(G_{t:t+n}, Z_{n+1} \mid s) = 0.$$

$\square$

**Assumption C.2** (**Coalition-Variance Monotonicity**). *Let $\widehat{V}(C, s)$ be an unbiased estimator of $v_\pi(s)$ from on-policy data restricted to agents in $C$. Assume $\mathrm{Var}(\widehat{V}(C, s))$ is nondecreasing in $|C|$ (larger coalitions $\Rightarrow$ richer interactions $\Rightarrow$ higher variance).*

**Remark C.3** (**Coalition-Variance Monotonicity in Game-Structured NAHT**). *Recall from Section 4.1 that $v_{C,s}^{v_s(\mathcal{M})} = v_s(\mathcal{M}) v_{C,s}^1$, and $v_{C,s}^1$ in a given state $s$ is isomorphic to $v_C^1$. Following Definition 2.1, we can express $v_{C,s}^{v_s(\mathcal{M})}$ as follows:*

$$v_{C,s}^{v_s(\mathcal{M})}(D) = \begin{cases} v_s(\mathcal{M}) & \text{if } C \subseteq D, \\ 0 & \text{otherwise}. \end{cases}$$

*As described in Definition 3.2, a basis game value $v_{C,s}^{v_s(\mathcal{M})}(D)$ can effectively evaluate the performance of a team of agents $C$[5] with an agent-type composition denoted by $\times_{i \in D} T_i$, where $T_i$ indicates the agent $i$'s type.*

*As the size of the coalition $C$ increases, it requires more experience for exploring agent-type compositions during agent interactions, so that the value of $V(C, s) := v_{C,s}^{v_s(\mathcal{M})}(\mathcal{M})$ can predict $v_s(\mathcal{M})$. It implies that as the coalition size increases, higher variance of the value prediction is incurred to get $v_s(\mathcal{M})$, due to the growing agent-type composition space. Similarly, if the sizes of two coalition are equal, their variances are equal.*

**Proposition C.4** (**Variance Monotonicity along the Coalition Enumeration**). *Fix $s \in \mathcal{S}$ and the sequence $(C_i, n_i)$ of Definition 4.4. Under the nonnegative-autocovariance condition of Lemma C.1 and Assumption C.2, the sequences*

$$i \mapsto \mathrm{Var}(\widehat{V}(C_i, s)) \quad \text{and} \quad i \mapsto \mathrm{Var}(G_{t:t+n_i} \mid s)$$

*are both nondecreasing in $i$. Moreover, $i \mapsto \mathrm{Var}(G_{t:t+n_i} \mid s)$ is strictly increasing at every index where $\mathrm{Var}(Z_{n_i+1} \mid s) > 0$.*

*Proof.* Since $n_i = n_{i-1} + 1$ is strictly increasing, Lemma C.1 yields nondecreasing variance in $i$, with strictness when the added step contributes nonzero variance. Assumption C.2 gives nondecreasing $\mathrm{Var}(\widehat{V}(C_i, s))$ as $|C_i|$ (hence $i$) increases. $\square$

**Remark C.5** (**Minimality of Horizon Schedule**). *Among all sequences $(\tilde{n}_i)$ with $\tilde{n}_1 = 1$, $\tilde{n}_i > \tilde{n}_{i-1}$, and $\tilde{n}_i \geq |C_i|$ for all $i$, the schedule in Definition 4.4 is pointwise minimal: $\tilde{n}_i \geq n_i$ for every $i$.*

---

[5]For any team of agents $D \supset C$, the basis game value would be the same as $v_{C,s}^{v_s(\mathcal{M})}(C)$, so it is ineffective to evaluate the performance of the $D$ with the larger size than $C$.

## D  COMPLETE MATHEMATICAL PROOFS

**Lemma 4.1.** *Given a fixed state $s \in \mathcal{S}$, each state-specific cooperative game's value $v_s \in \mathcal{G}(s)$ can be uniquely represented by: $v_s = \sum_{\emptyset \neq C \subseteq \mathcal{M}} k'_C \cdot v^z_{C,s}$, where $k'_C = \frac{k_C}{z}$ and $z \neq 0$.*

*Proof.* We can represent each $k_C$ in an equivalent form as $\frac{k_C \cdot z}{z}$, where $z \neq 0$. Substituting this term into the formula $v_s = \sum_{\emptyset \neq C \subseteq \mathcal{M}} k_C v^1_{C,s}$ in Definition 2.1, we can get the following formula such that:

$$v_s = \sum_{\emptyset \neq C \subseteq \mathcal{M}} \frac{k_C \cdot z}{z} \cdot v^1_{C,s}.$$

Since $v^z_{C,s} = z \cdot v^1_{C,s}$ by Definition 2.1, we have the following formula such that:

$$v_s = \sum_{\emptyset \neq C \subseteq \mathcal{M}} \frac{k_C}{z} \cdot v^z_{C,s}.$$

$\square$

**Proposition 4.2.** *For the class of superadditive games formed by a set of basis games $\{v^z_{C,s} | \emptyset \neq C \subseteq \mathcal{M}\}$, it holds that $k_C \geq 0$, for all $\emptyset \neq C \subseteq \mathcal{M}$.*

*Proof.* We first consider a cooperative game space $\mathcal{G}$ (a broader space of superadditive games). As per Dubey (1975), a game $v : \mathcal{G} \to \mathbb{R}_{\geq 0}$ belonging to $\mathcal{G}$ can be uniquely represented as follows:

$$v = \sum_{\emptyset \neq C \subseteq \tilde{N}} k_C v^1_C.$$

The analytic form of $k_C$ under $\mathcal{G}$ is represented as follows:

$$k_C = \sum_{T \subseteq C} (-1)^{|C|-|T|} v(T).$$

The condition for $v$ to be a superadditive game is as follows:

$$v(T_1 \cup T_2) \geq v(T_1) + v(T_2), \qquad T_1, T_2 \subseteq C.$$

Following the result from Dubey (1975), if the above condition holds, $k_C \geq 0$ has to be satisfied.[6]

We now extend the $\mathcal{G}$ to a state-specific cooperative game space $\mathcal{G}(s)$, for a fixed state $s \in \mathcal{S}$. Since the $\mathcal{G}(s)$ is isomorphic to a game space $\mathcal{G}$, the characteristics satisfied in the $\mathcal{G}$ also holds in the $\mathcal{G}(s)$ by Remark 3.1. Therefore, $v_s : \mathcal{G}(s) \to \mathbb{R}_{\geq 0}$ can be uniquely represented as $v_s = \sum_{\emptyset \neq C \subseteq \mathcal{M}} k_C v^1_{C,s}$. As we consider $\mathcal{G}(s)$ as a space of superadditive games, $k_C \geq 0$ should hold.

By Lemma 4.1, we have $v_s = \sum_{\emptyset \neq C \subseteq \mathcal{M}} k'_C \cdot v^z_{C,s}$, where $k'_C = \frac{k_C}{z}$ and $z \neq 0$. This implies that any set of basis games in the form $\{v^z_{C,s} | \emptyset \neq C \subseteq \mathcal{M}\}$ can form a state-specific cooperative game belonging to superadditive games. $\square$

**Proposition 4.6** (**Representation of Transformed Rewards**). *Given the condition $\sum_{i=1}^M \phi_i(R_t) = R_t$, the payoff allocation defined on rewards $R_t$, can be expressed as:*

$$\phi_i(R_t) := R_t - \sum_{j \neq i} \left( \phi_j(V(\mathcal{M}, s_t)) - \gamma \phi_j(V(\mathcal{M}, s_{t+1})) \right).$$

*Proof.* Recall that $\phi(\cdot) \in \mathbb{R}^M$ is a multidimensional linear transformation, where $M$ is the number of agents. We now express $\phi(R_t)$ by introducing the Efficiency axiom.

---

[6]For example, if $C = \{1, 2\}$, then $T = \emptyset, \{1\}, \{2\}, \{1, 2\}$. As a result, $k_C = v(\emptyset) - v(\{1\}) - v(\{2\}) + v(\{1, 2\})$, where $v(\emptyset) = 0$ by default. If $v$ is assumed to be a superadditive game, $k_C \geq 0$ has to hold.

To satisfy the Efficiency axiom such that $\sum_{i=1}^{M} \phi_i(V(\mathcal{M}, s_t)) = V(\mathcal{M}, s_t)$, it is reasonable to assume that $\sum_{i=1}^{M} \phi_i(R_t) = R_t$. In other words, each agent's value expansion can be expressed independently in its own $\phi_i(R_t)$, justified by Theorem 4.7.

Next, we aim to show how each $\phi_i(R_t)$ is expressed in terms of $\phi(V(\mathcal{M}, s_t))$.

It is not difficult to observe that for each agent $i \in \mathcal{M}$, we have

$$\phi_i(V(\mathcal{M}, s_t)) = \phi_i(R_t) + \gamma \phi_i(V(\mathcal{M}, s_{t+1})).$$

By the condition $\sum_{i=1}^{M} \phi_i(R_t) = R_t$, we can derive a formula such that:

$$\phi_i(R_t) = R_t - \sum_{j \neq i} \phi_j(R_t).$$

Since $\phi_j(R_t) = \phi_j(V(\mathcal{M}, s_t)) - \gamma \phi_j(V(\mathcal{M}, s_{t+1}))$, we can express $\phi_i(R_t)$ as follows:

$$\phi_i(R_t) := R_t - \sum_{j \neq i} \left( \phi_j(V(\mathcal{M}, s_t)) - \gamma \phi_j(V(\mathcal{M}, s_{t+1})) \right).$$

$\square$

**Theorem 4.8.** *The payoff allocation operator satisfying permutation-equivariance is a sufficient condition for the Symmetry axiom.*

*Proof.* Let $\mathcal{M} = \{1, \ldots, M\}$. Let $\mathcal{G}$ be the set of cooperative games $v : 2^{\mathcal{M}} \to \mathbb{R}$, and $S_C$ be a permutation group acting on $C \subseteq \mathcal{M}$.

Let $S_C$ act on an arbitrary $C \subseteq \mathcal{M}$ by exchanging agents:

$$(\sigma \cdot v)(C) := v(\sigma^{-1}(C)), \qquad \sigma \in S_{\mathcal{M}}, \ C \subseteq \mathcal{M}.$$

Let a payoff allocation operator $F : \mathcal{G} \to \mathbb{R}^M$ output payoffs $F(v) = (F_1(v), \ldots, F_n(v))$.

Let $S_{\mathcal{M}}$ act on $\mathbb{R}^M$ by permuting coordinates: $(\sigma \cdot x)_i := x_{\sigma^{-1}(i)}$.

$F$ is permutation-equivariant if

$$F(\sigma \cdot v) = \sigma \cdot F(v), \qquad \forall \sigma \in S_{\mathcal{M}}, \ v \in \mathcal{G}. \tag{20}$$

Recall that agents $i$ and $j$ are symmetric in $v$ if

$$v(C \cup \{i\}) = v(C \cup \{j\}) \quad \forall C \subseteq \mathcal{M} \setminus \{i, j\}.$$

Equivalently, the transposition $\tau = (i \ j)$ leaves the game value invariant:

$$\tau \cdot v = v. \tag{21}$$

**Next, we aim to show $F_i(v) = F_j(v)$ to prove the statement in the theorem.**

By Eqs. 20 and 21, we have

$$F(v) = F(\tau \cdot v) = \tau \cdot F(v).$$

Note that $\tau \cdot F(v)$ is just $F(v)$ with coordinates $i$ and $j$ swapped. Therefore, $F(v) = \tau \cdot F(v)$ implies the $i$-th and $j$-th entries are equal:

$$F_i(v) = F_j(v).$$

$\square$

**Theorem 4.10.** *Shapley Machine is an algorithm enforcing $V_i$ to fulfil Efficiency, Symmetry and Linearity, so the $V_i$ is the Shapley value for dynamic scenarios.*

*Proof.* By construction, the Shapley Machine enforces the individual value functions $V_i$ (payoff allocation functions) to satisfy the axioms of Efficiency, Symmetry, and Linearity. Each state-specific cooperative game space $\mathcal{G}(s)$ is isomorphic to the canonical cooperative game space $\mathcal{G}$ (Remark 3.1). Hence, properties of payoff allocation functions that hold on $\mathcal{G}$ also hold on $\mathcal{G}(s)$.

Now, recall that the Shapley value is uniquely characterized as the value function that satisfies Efficiency, Symmetry, and Additivity (Theorem 2.2). Since Linearity is a stronger condition than Additivity (indeed, setting all $\alpha_i = 1$ in the definition of Linearity yields Additivity), any payoff allocation function that is Efficient, Symmetric, and Linear must coincide with the Shapley value. Therefore, the individual value functions $V_i$ produced by the Shapley Machine are exactly the Shapley values for dynamic scenarios. $\square$

# E  EXPERIMENTAL DETAILS

## E.1  IMPLEMENTATION DETAILS

Our algorithm is built upon POAM and IPPO. All loss functions for training the encoder-decoder model, policy networks and value networks have been remained. Please refer to Appendix B.3 for details. For conciseness, we only list the novel loss functions proposed in this paper as below.

### E.1.1  SHAPLEY MACHINE

We now describe the details about implementing our proposed algorithm, referred to as Shapley Machine. In general, our algorithm is established based on the base algorithms POAM and IPPO, with modification to fulfil all the three axioms of Shapley value: Efficiency, Linearity and Symmetry. Since Symmetry has been implemented by structuring the inputs as shown in Remark 4.9, we only need to fulfil Efficiency and Linearity as follows.

**Implementing the Linearity Axiom.** In general, both POAM and IPPO have implemented TD($\lambda$), which does not strictly conform to the principle of the Linearity axiom. To this end, we change the TD($\lambda$) to TTD($\lambda$), where the number of n-step return components is equal to the number of the non-empty coalitions in theory. Note that in some scenarios the episode length is smaller than the number of non-empty coalitions. In these cases, the TTD($\lambda$) can be reduced to the TD($\lambda$) for the finite-horizon tasks with the episode length as $T$, equivalently, the TTD($\lambda$) with $m = T$. Alternatively, we can select a value of $m$ for each task based on extra conditions, which is left for the future work. In this paper, we set $m = T$ for the scenarios 8v9 and 10v11. For the 8v9 scenario, the episode length as 120 is not too far from the number of non-empty coalitions as 127. For the 10v11 scenario, the episode length is 150, while the number of non-empty coalitions is 511.

**Implementing the Efficiency Axiom.** By introducing the partial observation in Dec-POMDP, the Eq. 10 we have derived the condition for realizing the Efficiency axiom is transformed as follows:

$$R_{t,i} = R_t - \sum_{j \neq i} \left( V_j(h_{t,j}) - \gamma V_j(h_{t+1,j}) \right),$$

where $V_j(h_{t,j})$ indicates an agent $j$'s non-parameterized individual value. During the practical training procedure, the $V_j(h_{t,j})$ generated by the individual value network could be severely inaccurate in the beginning, which may result in the instability of learning. To mitigate this issue, we add an extra coefficient $\alpha \in (0, 1)$ to the term $\sum_{j \neq i} \left( V_j(h_{t,j}) - \gamma V_j(h_{t+1,j}) \right)$, such that:

$$R_{t,i} = R_t - \alpha \sum_{j \neq i} \left( V_j(h_{t,j}) - \gamma V_j(h_{t+1,j}) \right). \tag{22}$$

This coefficient $\alpha$ can be either manually set up as a fixed value, or implemented by a scheduler starting from 0 to some preset upper limit. The $R_t$ in $L_\theta(h_{t,i}, a_{t,i})$ and $L_{\theta^c}(h_{t,i})$ is replaced by the above $R_{t,i}$. To clarify this change, the two new losses are expressed as: $\hat{L}_\theta(h_{t,i}, a_{t,i})$ and $\hat{L}_{\theta^c}(h_{t,i})$.

Furthermore, it is needed to search the underlying factorization scheme of the transition function in the Dec-POMDP, according to Theorem 4.7. To implement this requirement, we need to fulfil the following condition:

$$\sum_{i=1}^{M} V_i(h_{t,i}) = V(\mathcal{M}, h_t). \tag{23}$$

The above equality is implemented as a regularization term during training.

To maintain consistency with the critic losses, we consider to use the $\lambda$-return denoted by $\hat{G}_t^\lambda$ to represent $V(\mathcal{M}, h_t)$. In turn, the above equation can be expressed as:

$$V(\mathcal{M}, h_t) = \mathbb{E}_\pi \left[ G_t^\lambda \middle| h_t \right],$$

where each $\hat{G}_{t:t+n}$ contributing to the $\hat{G}_t^\lambda$ represented in TTD($\lambda$) is expressed as follows:

$$\hat{G}_{t:t+n} = R_t + \gamma R_{t+1} + \cdots + \gamma^{n-1} R_{t+n-1} + \gamma^n V(\mathcal{M}, h_{t+n})$$

$$= R_t + \gamma R_{t+1} + \cdots + \gamma^{n-1} R_{t+n-1} + \gamma^n \sum_{i=1}^{M} V_i(h_{t+n,i}) \quad \text{(By Eq. 23)}.$$

Substituting the above formula into Eq. 23, we can obtain the regularization term referred to as the **efficiency loss** for one timestep $t$, as follows:

$$L_{\theta^c}^e(h_{t,i}) = \frac{1}{2} \left( \hat{G}_t^\lambda - \sum_{i=1}^{M} V_i^{\theta^c}(h_{t,i}) \right)^2.$$

In summary, the total loss function of Shapley Machine for POAM (SM-POAM) is as follows:

$$L_{\text{SM-POAM}} = \frac{1}{T} \sum_{t=1}^{T} \left( \sum_{i \in \mathcal{C}_t} \hat{L}_\theta(h_{t,i}, a_{t,i}) + \beta_1 \sum_{i \in \mathcal{M}} \hat{L}_{\theta^c}(h_{t,i}) + \beta_2 L_{\theta^c}^e(h_{t,i}) + L_{\theta^e, \theta^o, \theta^a}(h_{t,i}, o_{t,-i}, a_{t,-i}) \right),$$

where $T$ is the episode length; $\beta_1, \beta_2 \in (0, 1)$ are two coefficients to control the importance of the two losses; as well as $\mathcal{C}_t$ indicates the controlled agent set at timestep $t$ and $\mathcal{M}$ indicates the ad hoc team following the convention in Wang et al. (2025).

Similarly, the total loss function of Shapley Machine for IPPO (SM-IPPO) is as follows:

$$L_{\text{SM-IPPO}} = \frac{1}{T} \sum_{t=1}^{T} \left( \sum_{i \in \mathcal{C}_t} \hat{L}_\theta(h_{t,i}, a_{t,i}) + \beta_1 \sum_{i \in \mathcal{M}} \hat{L}_{\theta^c}(h_{t,i}) + \beta_2 L_{\theta^c}^e(h_{t,i}) \right).$$

**Implementation of $k'_C > 0$.** As mentioned in Section 4.1, it is necessary to fulfil $k'_C > 0$ for reaching cooperation, which can be implemented following the convention of TD($\lambda$) in RL (Sutton, 1988). Specifically, the weightings for $m$ basis games $(k'_{C_1}, k'_{C_2}, \cdots, k'_{C_m})$ are generated using a geometric distribution $P_\lambda$ with the parameter $0 < \lambda < 1$, such that $k'_{C_n} = P_\lambda(n)$, resulting a tuple $((1-\lambda), (1-\lambda)\lambda, \cdots, (1-\lambda)\lambda^{m-1}, \lambda^m)$. With this condition, Eq. 8 becomes the TD error of the well-known **truncated TD($\lambda$) prediction**, shortened as **TTD($\lambda$)** (Cichosz, 1994).

**Partial Observations.** In practice, a controlled agent is only able to receive an observation, following the settings of Dec-POMDPs. Therefore, an agent is required to infer the state of the environment as an individual hidden state, through the history of observations. To this end, the policy and individual networks are realized by recurrent neural networks (RNNs) (e.g., GRUs (Chung et al., 2014)), where observations or representations transformed from observations are as inputs. In implementation, we use two base algorithms satisfying the conditions of partial observations: POAM (Wang et al., 2025) and IPPO (De Witt et al., 2020) to realize Shapley Machine, referred to as: **SM-POAM** and **SM-IPPO**, respectively.

**Implementation of the Symmetry Axiom.** It can be observed that both POAM and IPPO are implemented by the sharing parameters, with agent ID to differentiate agent identities. Given an agent

ID, its input is either individual observations or individual observations + teammate embeddings, where the teammate embeddings are transformed from individual observations. As per Remark 4.9, **the structures of SM-POAM and SM-IPPO satisfies the Symmetry axiom**.

**Estimation of Uncontrolled Agent Individual Values.** We maintain all possible controlled agent policy and individual values networks as implemented by Wang et al. (2025), given that the maximum number of controlled agents is known in the experimental settings of the NAHT, but only part of controlled agents in an episode can make decision. Thanks to the sharing parameters technique, this can be simply implemented by maintaining one policy or individual value network with a tuple of agent IDs across all possible controlled agents. During the training phase, the uncontrolled agent individual values are also required to implement the Efficiency axiom, as informed in Eq. 11, but they are unknown to controlled agents. To this end, the uncontrolled agent individual values are approximated by the maintained individual values of the controlled agents which are not activated.

### E.1.2  POAM AND IPPO

The implementations of POAM and IPPO have been detailed in Appendix B.3.

In summary, the total loss function of POAM is as follows:

$$L_{\text{POAM}} = \frac{1}{T} \sum_{t=1}^{T} \left( \sum_{i \in \mathcal{C}_t} L_\theta(h_{t,i}, a_{t,i}) + \beta_1 \sum_{i \in \mathcal{M}} L_{\theta^c}(h_{t,i}) + L_{\theta^e, \theta^o, \theta^a}(h_{t,i}, o_{t,-i}, a_{t,-i}) \right).$$

The total loss function of IPPO is as follows:

$$L_{\text{IPPO}} = \frac{1}{T} \sum_{t=1}^{T} \left( \sum_{i \in \mathcal{C}_t} L_\theta(h_{t,i}, a_{t,i}) + \beta_1 \sum_{i \in \mathcal{M}} L_{\theta^c}(h_{t,i}) \right).$$

### E.1.3  BANZHAF MACHINE

The overall loss function and implementation of the Banzhaf Machine closely follow those of POAM and IPPO. The only distinction is that Banzhaf Machine employs TTD($\lambda$) in place of the TD($\lambda$) used in POAM and IPPO.

## E.2  EXPERIMENTAL DOMAINS

We now briefly introduce the experimental domains for running experiments. If one would like to know more about details, please refer to Wang et al. (2025).

### E.2.1  MPE PREDATOR PREY (MPE-PP)

The MPE Predator Prey (MPE-PP) environment is a predator-prey task implemented within the Multi-Agent Particle Environment (MPE) framework. It simulates interactions within a two-dimensional space populated by two static obstacles, where three pursuer agents must cooperate to capture a single adversarial evader. A successful capture is defined as at least two pursuers simultaneously colliding with the evader, upon which the pursuers receive a positive reward of +1. If the capture is unsuccessful, no reward is granted. This environment is designed to test the ability of agents to coordinate under spatial and dynamic constraints.

### E.2.2  THE STARCRAFT MULTI-AGENT CHALLENGE (SMAC)

The StarCraft Multi-Agent Challenge (SMAC) serves as a benchmark suite for evaluating MARL algorithms in partially observable, cooperative settings. Built atop the StarCraft II game engine, SMAC presents a variety of micromanagement tasks where each agent (e.g., a Marine or Stalker) operates based on limited local observations and must coordinate actions with teammates to overcome enemy units. In this work, we focus on four specific SMAC scenarios: **5v6**: Five allied Marines versus six enemy Marines, **8v9**: Eight allied Marines versus nine enemy Marines, **10v11**: Ten allied Marines versus eleven enemy Marines, **3s5z**: Three allied Stalkers versus five enemy Zealots. At each timestep, agents receive a shaped reward proportional to the damage they inflict on opponents, along

with bonus rewards of 10 points for each enemy defeated and 200 points for achieving overall victory by eliminating all adversaries. The total return is normalized so that the maximum achievable return in each scenario is 20. The action space in SMAC is discrete, enabling each agent to choose actions such as attacking a particular enemy, moving in a specific direction, or remaining idle. Notably, the variation in the number and type of agents and opponents across tasks results in scenario-specific observation and action space dimensionalities, thereby introducing further diversity and complexity for algorithmic evaluation. The length of an episode varies across different scenarios: MPE-PP with $T = 100$, 3sv5z with $T = 250$, 5v6 with $T = 70$, 8v9 with $T = 120$ and 10v11 with $T = 150$.

### E.3 EVALUATION METRICS

For in-distribution evaluation, the mean return (or winrate) is computed over $E$ randomly sampled episodes. In each episode, we form a joint policy $\pi^{(\mathcal{M})}$ by sampling $N$ agents uniformly from $\mathcal{C}$ and the remaining $M - N$ agents from $\mathcal{U}$. For the out-of-distribution (OOD) evaluation, the M–N score (Wang et al., 2025) is applied. Given a set of controllable agents $\mathcal{C}$ and uncontrolled agents $\mathcal{U}$, the M–N score measures how well mixed teams of these agents cooperate in the NAHT setting. The score is computed exhaustively by varying the number of controllable agents included in the team. Specifically, for each $N \in \{1, \ldots, M - 1\}$, we form a joint policy $\pi^{(\mathcal{M})}$ by sampling $N$ agents uniformly from $\mathcal{C}$ and the remaining $M - N$ agents from $\mathcal{U}$. The mixed team is then evaluated on the task for $E$ episodes. This procedure yields $(M - 1) \cdot E$ returns (or winrates) in total, whose average defines the M–N score. For the OOD evaluation on unseen conventions, the M–N score is computed by averaging the per-convention M–N scores across 4 other conventions for each agent type. For the OOD evaluation on unseen agent types, the per-convention M–N score is computed by averaging the per-convention M–N score across 5 conventions for each unseen agent type. Each convention of uncontrolled agents is a random seed that pretrains them. All results are obtained by first computing each metric with 128 episodes and then averaging these per-seed metrics across 5 random seeds, with 95% confidence intervals reported.

### E.4 EXPERIMENTAL SETTINGS

The training procedure of the n-agent ad hoc teamwork (NAHT) process is briefly introduced here. For more details, please refer to Wang et al. (2025). For each scenario (e.g. MPE, 3sv5z, 5v6, 8v9 and 10v11), there are five groups of pretrained agents (e.g. IQL, MAPPO, VDN, QMIX and IPPO) acting as the uncontrolled agents. For each episode evaluation, the number of uncontrolled agents $M - N$ is sampled, and then a group of pretrained agents is sampled. Given that each task specifies a fixed total number of agents $M$, the number of controlled agents is $N$. **Note that this is still a special case of openness. For a varying number of controlled agents, the number of uncontrolled agents is also varied in correspondence, constrained by the total number of agents related to the task specifications.** The distribution for all sampling procedures are modelled as the multinational distribution with no replacement. Each uncontrolled agent executes the greedy policy in both training and testing procedures. In contrast, each controlled agent executes the on-policy sampling via the parameterized policy during the training procedure, while the greedy policy during the testing procedure.

### E.5 HYPERPARAMETER SETTINGS

Since our algorithm is established based on POAM and IPPO, most hyperparameter settings follow that in Wang et al. (2025). First, the actors and critics are implemented in recurrent neural networks, with full parameter sharing. Specifically, they are implemented by two fully connected layers followed by a GRU layer and an output layer. Each layer has 64 dimensions with a ReLU activation function, and layer normalization is applied. The encoder-decoder networks for inferring agent characteristics are also implemented in parameter sharing. The encoder is implemented by a GRU layer, followed by a fully connected layer with a ReLU activation function and an output layer. The decoder is implemented by two fully connected layers with ReLU activation functions, followed by an output layer. Adam Optimizer is used to train all models. The detailed hyperparameter for experiments is shown in Tables 1 and 2. For the scenarios such as MPE, 3sv5z and 5v6, the values of $m$ are set as the number of non-empty coalitions. For the scenarios such as 8v9 and 10v11, the values of $m$ is simply set as the length of an episode (see Appendix E.1 for more details). Note that the term

$\sum_{j\neq i}(V_j(s_t) - \gamma V_j(s_{t+1}))$ in shaped rewards is standardised to match the scales of standardised rewards. To stabilize learning in large-scale scenarios, we add a value loss clip to Shapley Machine and Banzhaf Machine for POAM on both 8v9 and 10v11, and for IPPO on 10v11 only.

Table 1: Key hyperparameters of Shapley and Banzhaf Machine: $\lambda$ indicates the parameter for the weighting functions of TTD($\lambda$), $m$ indicates the number of basis games, $\alpha$ controls the importance of the term $\sum_{j\neq i}(V_j(s_t) - \gamma V_j(s_{t+1}))$ in Eq. 22, $\beta_1$ controls the importance of the critic loss $\sum_{i\in\mathcal{M}}\hat{L}_{\theta^c}(h_{t,i})$, and $\beta_2$ controls the importance of the efficiency loss $L_{\theta^c}^e(h_{t,i})$.

| Algorithm | Task | $\lambda$ | $m$ | $\alpha$ | $\beta_1$ | $\beta_2$ |
|---|---|---|---|---|---|---|
| Shapley Machine | MPE-PP | 0.85 | 7 | 0.01 | 0.5 | 0.01 |
| | 3sv5z | 0.85 | 7 | 0.01 | 0.5 | 0.01 |
| | 5v6 | 0.85 | 31 | 0.01 | 0.5 | 0.001 |
| | 8v9 | 0.95 | 120 | 0.01 | 0.5 | 0.001 |
| | 10v11 | 0.95 | 150 | 0.01 | 0.5 | 0.001 |
| Banzhaf Machine | MPE-PP | 0.85 | 7 | N/A | 0.5 | N/A |
| | 3sv5z | 0.85 | 7 | N/A | 0.5 | N/A |
| | 5v6 | 0.85 | 31 | N/A | 0.5 | N/A |
| | 8v9 | 0.95 | 120 | N/A | 0.5 | N/A |
| | 10v11 | 0.95 | 150 | N/A | 0.5 | N/A |

Table 2: Common hyperparameters of RL settings for both IPPO and POAM.

| Hyperparameter | Value |
|---|---|
| LR | 0.0005 |
| Epochs | 5 |
| Minibatches | 1 |
| Buffer size | 256 |
| Entropy coefficient | 0.05 |
| Clip | 0.2 |
| ED LR | 0.0005 |
| ED epochs | 1 |
| ED Minibatches | 1 |
| Optim_alpha (Adam) | 0.99 |
| Optim_eps (Adam) | 0.00001 |
| Use_obs_norm | True |
| Use_orthogonal_init | True |
| Use_adv_std | True |
| Standardise_rewards | True |
| num_parallel_envs | 8 |

### E.6 COMPUTATIONAL RESOURCES

All experiments are conducted on Intel Xeon Gold 6230 CPUs and Nvidia V100-SXM2 GPUs. Each experiment run on MPE takes approximately 7 hours, utilizing 20 CPU cores and 1 GPU. Each experiment run on SMAC takes between 8 and 19 hours, utilizing 30 CPU cores and 1 GPU. All experiments are trained with 20M timesteps. The post-training evaluation for each scenario takes between 6 and 16 hours, utilizing 8 CPU cores and 1 GPU.

## F ADDITIONAL EXPERIMENTS

### F.1 EMPIRICAL EVIDENCE FOR THE STRENGTH OF THE AXIOMATIC FRAMEWORK

**We now discuss the phenomenon observed from the performance under stochastic policy exploration during training.** Discussing this phenomenon is meaningful. When a machine learning

algorithm is deployed and has to adapt to real-world environments through online learning due to the mismatch between simulators and real-world environments, it is expected that random exploration diminishes rapidly over time to avoid the damage on real-world environments (e.g. physical systems).

Recall that the PPO family algorithms implement exploration by the natural stochastic policy sampling from the learned policy distribution and maximizing policy entropy. As seen from Figure 7, either Shapley- or Banzhaf-based algorithms can outperform their base algorithm counterparts with a large margin. The reasons are as follows. As shown in Figure 8, policy entropy decreases rapidly over the course of training (except for the IPPO implementation on 3sv5z, which underperforms due to incompatibility between IPPO and the task). This indicates that the learned policies become more deterministic than those of the base algorithms, despite all methods being optimized with an entropy-maximization objective. A possible explanation for this phenomenon is illustrated in Figure 9: the critic loss of Shapley and Banzhaf Machine decreases faster than that of their base algorithm counterparts. **This provides evidence for the advantage of structured individual value functions under our proposed axiomatic framework**.

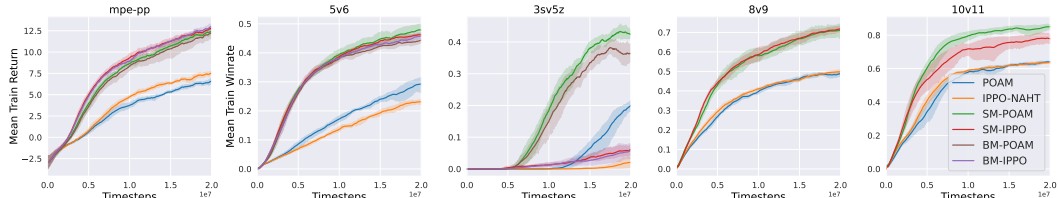

Figure 7: Performance under the stochastic exploration policy during training in NAHT.

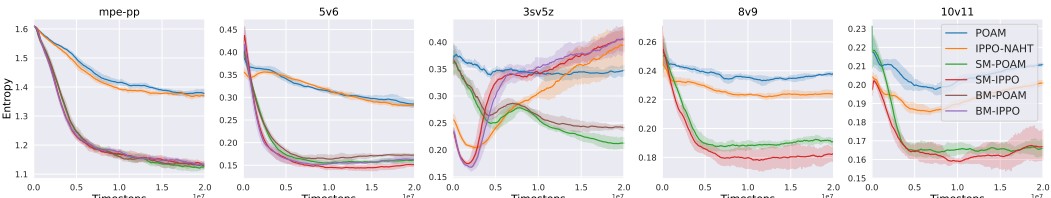

Figure 8: Policy entropy across scenarios during training in NAHT.

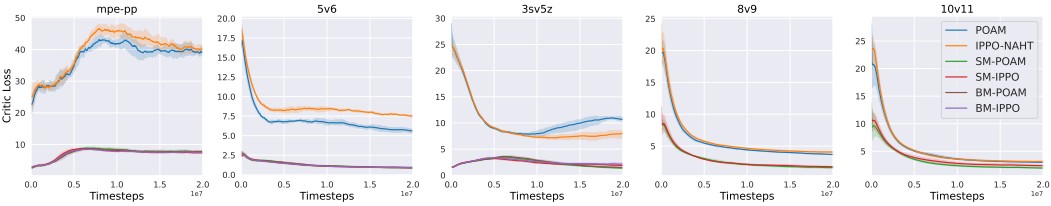

Figure 9: Critic loss across scenarios during training in NAHT.

## F.2 ANALYZING EFFECTS OF EFFICIENCY AXIOMS

We now demonstrate the association between the changing of the mean shaping rewards and the response of corresponding testing returns. As shown in Figure 12, the shaped rewards increases in the beginning and then gradually reaches a plateau. The efficiency loss also exhibits the similar phenomenon. As shown in Figure 13, it can be observed that when instability happens during training as evidenced by the sudden changes of test returns, the shaped rewards can consistently manifest this situation (as highlighted in red vertical dashed lines). This feature is crucial in the NAHT, since it could be a frequent and typical case when an unseen agent appears in the environment, which may perturb the training stability. **This justifies the effectiveness of the shaping rewards $R_{t,i}$ we propose to fulfil the Efficiency axiom. More importantly, this highlights the importance and necessity of the Efficiency axiom to perceive and measure the changing of environments**.

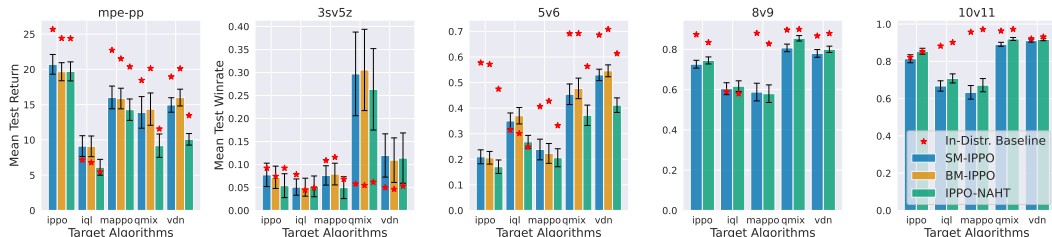

(a) Evaluation on unseen conventions (identical algorithms with different random seeds). The star denotes the in-convention baseline, and the histograms show performance on unseen conventions with error bars.

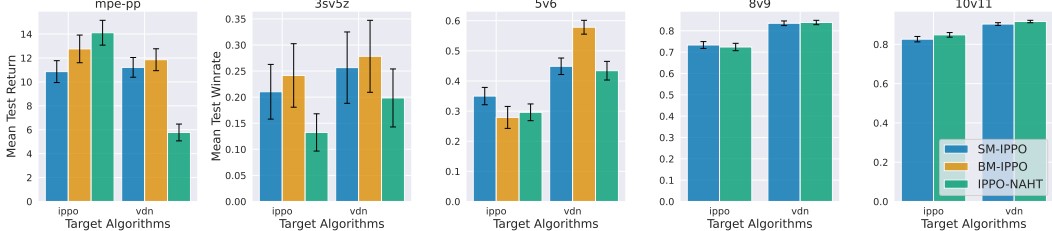

(b) Evaluation on unseen agent types. Train set: MAPPO, QMIX, IQL; test set: IPPO, VDN.

Figure 10: Out-of-distribution (OOD) evaluation after training, assessing unseen conventions and unseen agent types during training, between learned agents (IPPO-NAHT, SM-IPPO and BM-IPPO) and uncontrolled agent types (IPPO, VDN, IQL, QMIX and MAPPO). The performance is evaluated by averaging all pairings of N controlled and $M - N$ uncontrolled agents and their corresponding random seeds, called M–N score (Wang et al., 2025).

### F.3 EXTRA RESULTS OF OOD EVALUATION

We also conducted experiments on evaluating the OOD performance of Shapley Machine and Banzhaf Machine on the base algorithm IPPO (IPPO-NAHT). As seen from Figure 10a, Banzhaf Machine is competitive to Shapley Machine on dealing with unseen conventions. Although it seems the result is opposite to the conclusion drawn from the POAM related results, the underlying reason could be the general weaker performance of IPPO than POAM due to the lack of agent modelling, as shown in Figure 14. This could inhibit the full capability of Shapley Machine. As seen from Figure 10b, the general performance of Banzhaf Machine is better than Shapley Machine. This is consistent with the conclusion drawn from the POAM related results. In Figure 11, we show additional comparisons between algorithms: POAM vs. IPPO, SM-POAM vs. SM-IPPO and BM-POAM vs. BM-IPPO. **We find no strong evidence that the axioms are sensitive to the choice of base algorithm.**

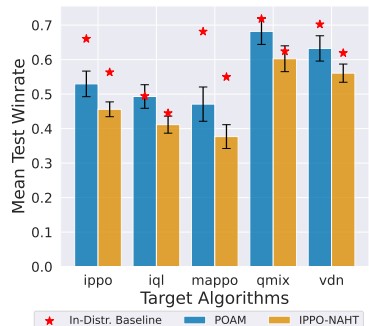

Figure 14: Average evaluation performance on unseen conventions across 4 SMAC tasks. The star denotes the in-convention baseline, and the histograms denotes the unseen conventions.

### F.4 POTENTIAL INSTABILITY OF SHAPLEY MACHINE FOR LARGE-SCALE SCENARIOS

During our repeated trials, we observe that the performance of Shapley Machine is not stable in large-scale scenarios (e.g. 8v9 and 10v11), compared with in small-scale scenarios (e.g. MPE-PP, 3sv5z and 5v6). We now analyze the potential reason behind this phenomenon. As observed from Figures 15 and 16, it can be confirmed that **the instability of shaped rewards is the key reason to cause the instability of learning procedure for large-scale scenarios**. According to the functionality of shaping rewards discussed in Appendix F.2, **we hypothesize that the instability could be caused by the unstable changing of value difference in the shaping reward:** $\sum_{j \neq i} (V_j(s_t) - \gamma V_j(s_{t+1}))$. Specifically, when the number of agents increases, the accumulating prediction error of individual value functions will be amplified.

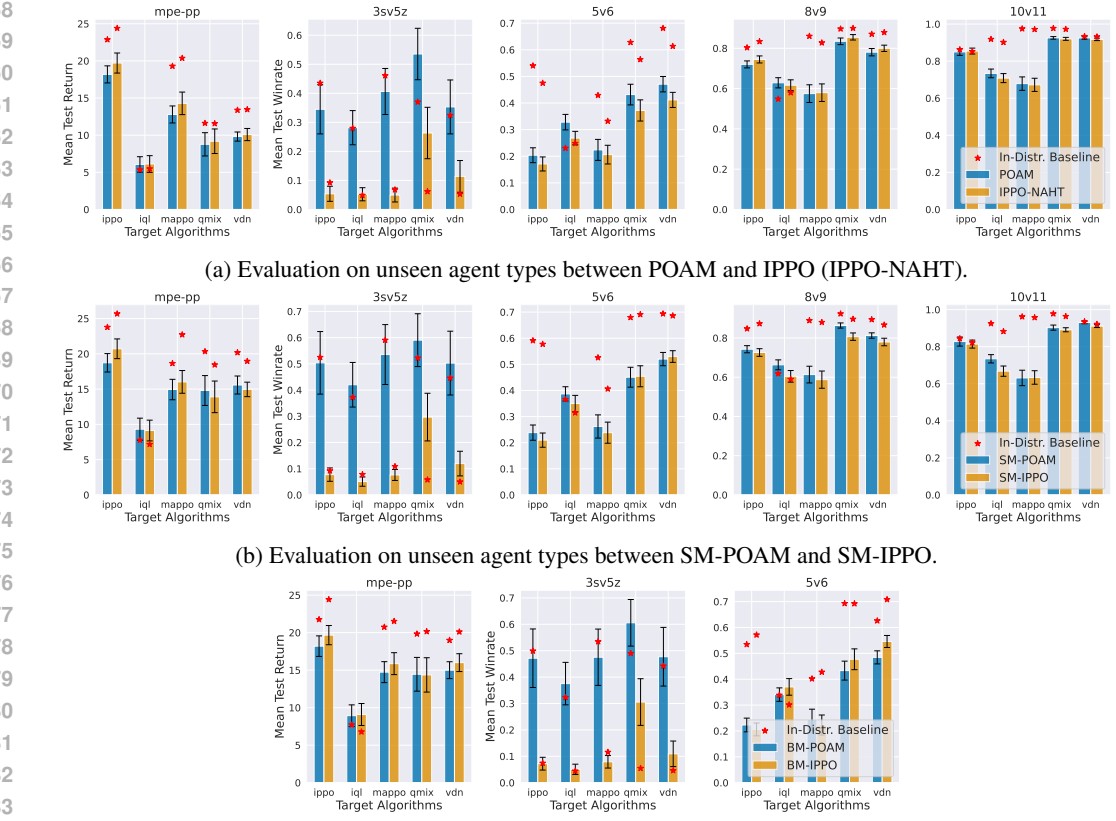

(a) Evaluation on unseen agent types between POAM and IPPO (IPPO-NAHT).

(b) Evaluation on unseen agent types between SM-POAM and SM-IPPO.

(c) Evaluation on unseen agent types between BM-POAM and BM-IPPO. Since the implementation of Banzhaf Machine is equivalent to its counterpart baseline algorithms for 8v9 and 10v11 due to approximating the number of non-empty basis games as the episode length, we ignore their evaluation here.

Figure 11: Out-of-distribution (OOD) evaluation after training, assessing unseen conventions during training, between learned agents (IPPO-NAHT, SM-IPPO, and BM-IPPO) and uncontrolled agent types (IPPO, VDN, IQL, QMIX and MAPPO). The performance is evaluated by averaging all pairings of N controlled and $M - N$ uncontrolled agents and their corresponding random seeds, called M–N score (Wang et al., 2025).

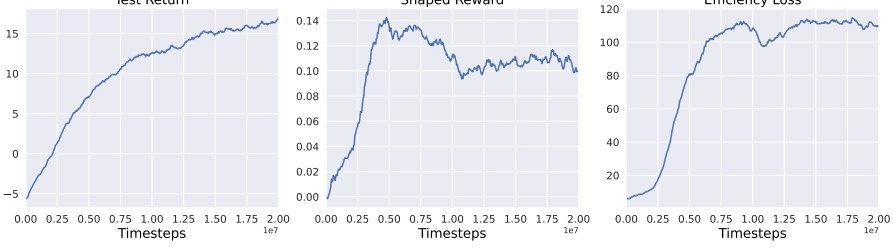

Figure 12: One run of MPE to show the variation of test return, shaped reward and efficiency loss.

To mitigate the above issue, we have added the value loss clip to stabilize the learning procedure of individual value functions for the large-scale scenarios such as 8v9 and 10v11. As seen from Figure 3, even with the value loss clip the instability of SM-IPPO is still out of control. This could be the cascading effect caused by the lack of agent modelling in contrast to SM-POAM, the instability of which is far beyond the capability of value loss clip. On the other hand, this reflects **the value of agent modelling in POAM for tackling the large-scale multi-agent scenarios**.

As seen from Figure 17, the learning procedures with the value loss clip for the SM-POAM apparently perform more stably than those without. This verifies our initial hypothesis. Due to this strategy will

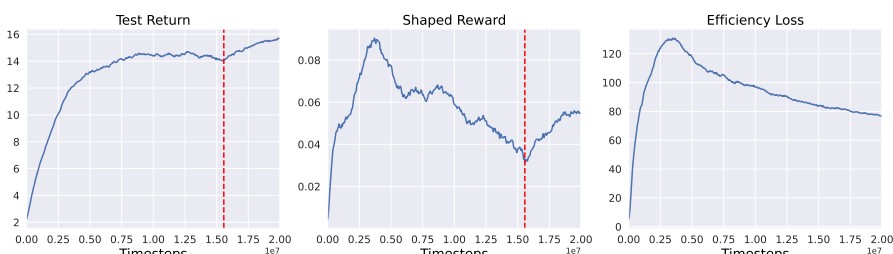

Figure 13: One run of 5v6 to show the variation of test return, shaped reward and efficiency loss.

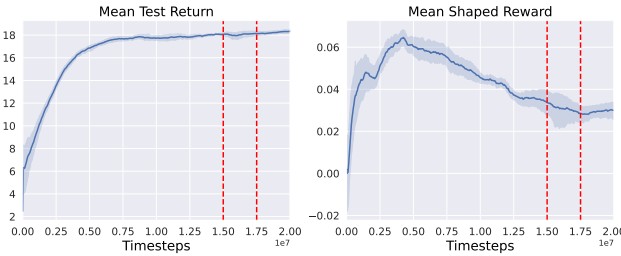

Figure 15: A case of 8v9 with SM-POAM to show the instability of learning progress via the mean shaped rewards. The range bounded by two red dashed lines shows the fluctuation of the learning process.

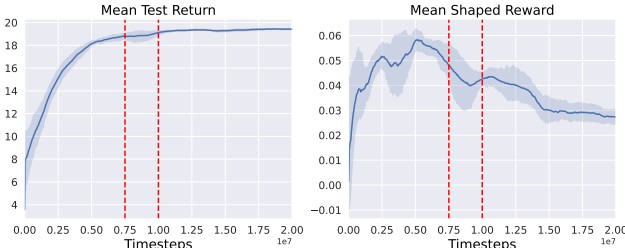

Figure 16: A case of 10v11 with SM-POAM to show the instability of learning progress via the mean shaped rewards. The range bounded by two red dashed lines shows the fluctuation of the learning procedure.

slow down and even hinder the performance for the other three scenarios, we have not posted this as a common strategy. **We believe this deserves further investigation in the future before any claims can be made about its general performance.**

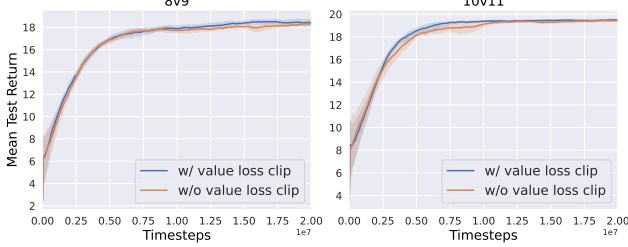

Figure 17: Comparison between the learning procedure of SM-POAM with and without the value loss clip in the 8v9 and 10v11 scenarios.

## F.5 OOD Evaluation Across Different Numbers of Controlled Agents

We report the results on unseen conventions and agent types across each specific number of controlled agents $N$ in Figures 18–21, by fixing the N value in the M–N score.

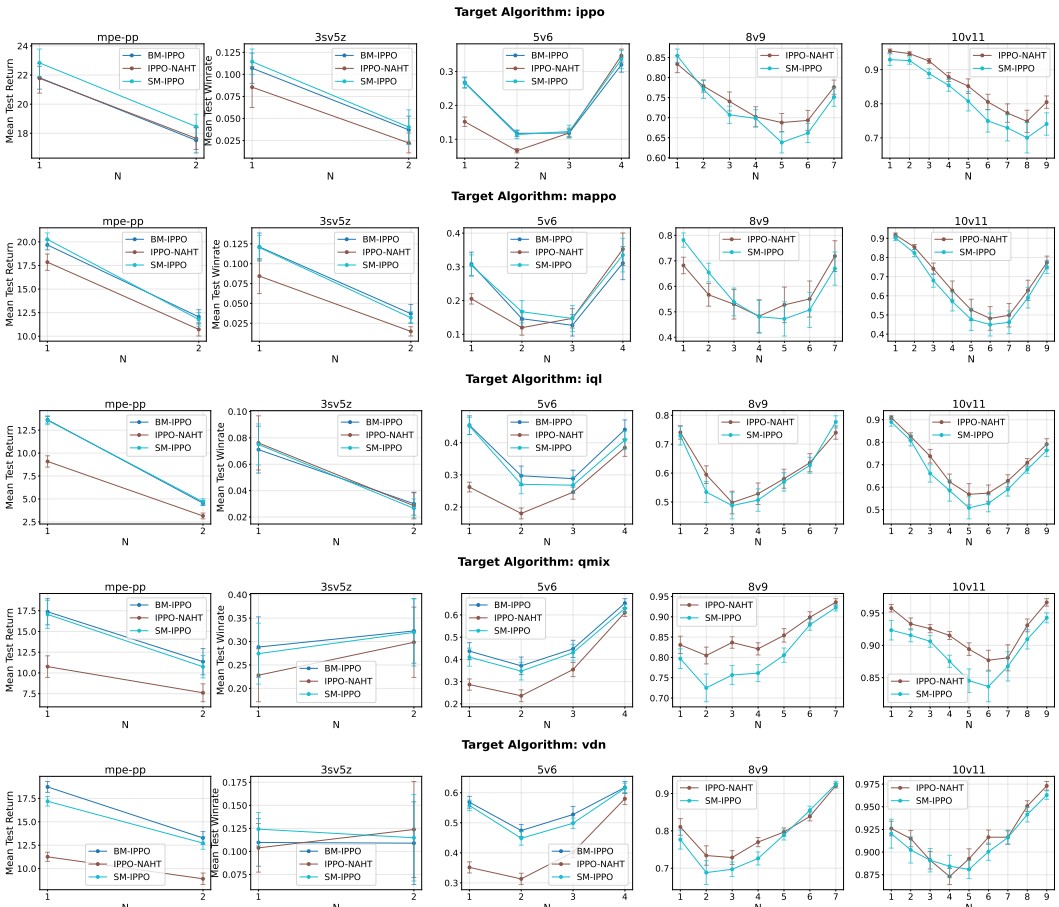

Figure 18: Out-of-distribution (OOD) evaluation after training, assessing unseen conventions during training, between learned agents (IPPO-NAHT, SM-IPPO and BM-IPPO) and uncontrolled agent types (IPPO, VDN, IQL, QMIX and MAPPO).

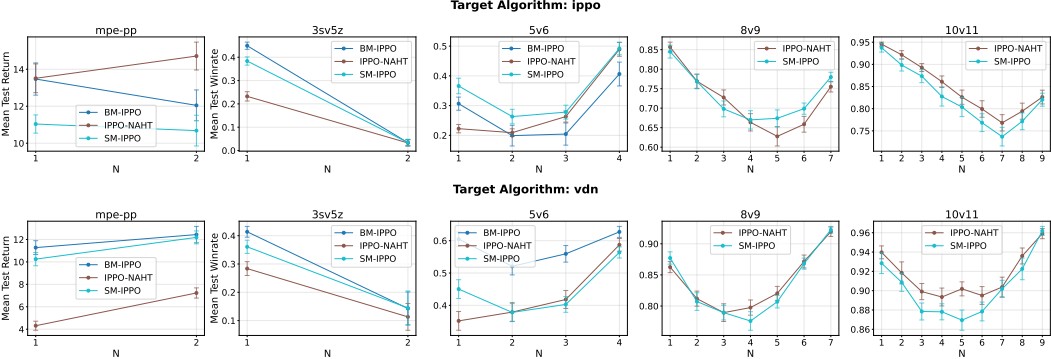

Figure 19: Out-of-distribution (OOD) evaluation after training, assessing unseen agent types during training, between learned agents (IPPO-NAHT, SM-IPPO and BM-IPPO) and unseen uncontrolled agent types (IPPO and VDN).

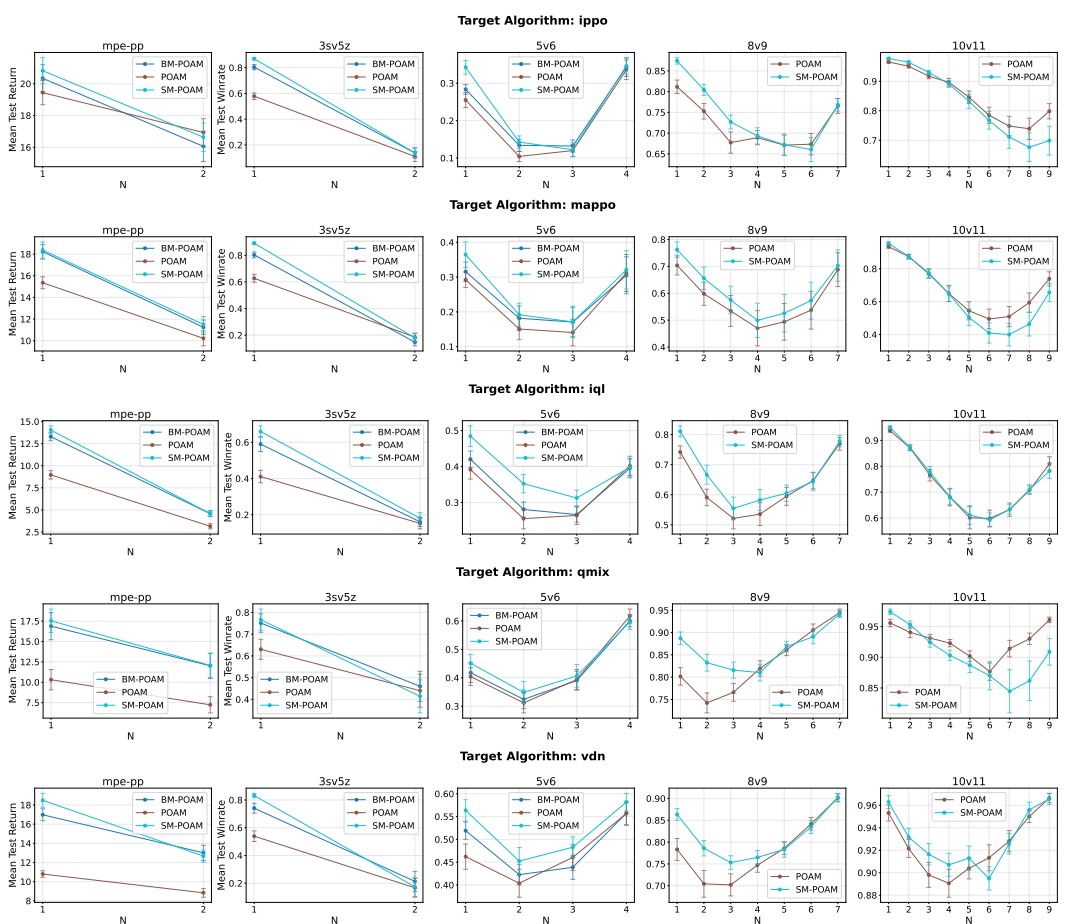

Figure 20: Out-of-distribution (OOD) evaluation after training, assessing unseen conventions during training, between learned agents (POAM, SM-POAM, and BM-POAM) and uncontrolled agent types (IPPO, VDN, IQL, QMIX and MAPPO).

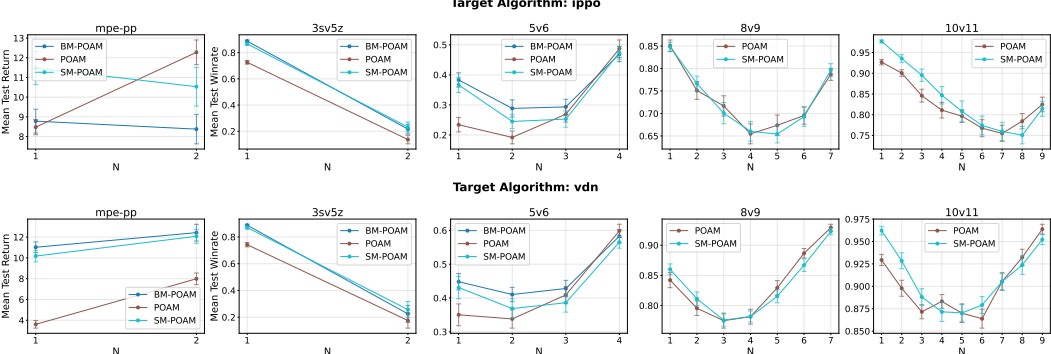

Figure 21: Out-of-distribution (OOD) evaluation after training, assessing unseen agent types during training, between learned agents (POAM, SM-POAM and BM-POAM) and unseen uncontrolled agent types (IPPO and VDN).

