# OpenReview forum: "An Axiomatic Framework for N-Agent Ad Hoc Teamwork: From Shapley Axioms to Learning"
_ICLR.cc/2026/Conference — Submitted to ICLR 2026_

### Official Review · Reviewer_yG89 · 2025-10-23

**Soundness:** 3
**Presentation:** 2
**Contribution:** 1
**Rating:** 4
**Confidence:** 2

**Summary:**

This paper introduces an axiomatic framework for ad hoc teamwork (NAHT), a setting where the payoff structure of the task is cooperative but only a subset of the other agents are controllable. By enforcing individual constraints on the value functions of each agent, the authors are able to induce the Shapley axioms of efficiency, linearity and symmetry. At the same time, this paper explores the results of enforcing only the symmetry and linearity constraints which they term as Banzhaf machine.

**Strengths:**

Overall the mathematical axioms and propositions seem sound and correct to me, in such way the theoretical foundations of the paper are very solid. The main strengths of this paper in my opinion are:

1. **Theoretical contributions:** The paper cleanly maps Shapley axioms to training constraints (reward shaping + symmetry + TTD($\lambda$)), with a formal statement that enforcing all three recovers Shapley values in dynamic settings (Theorem 4.10).

2. **Empirical gains on strong Benchmarks:** The proposed  Banzhaf (BM) and Shapley (SM) machines show improved returns on two difficult tasks (MPE and SMAC) and across a variety of multi-agent reinforcement learning algorithms.

3. **Reproducibility details:** Hyperparameters ($\lambda$, m, $\alpha$, loss weights), episode lengths, and compute budgets are reported.

**Weaknesses:**

The main weakness is that the paper is very difficult to understand, building very little on intuitions and quickly skimming over the required background. As someone who is not an expert in this field, I had to re-read the paper several times before getting a notion of what the authors where trying to do and why. More specifically the following are my issues with this submission:

1. **Problem motivation**: The authors make no statement about the motivation of the problem or what they are attempting to achieve concretely. Although previous approaches rely on heuristics, and this paper claims to construct a theoretical grounding for ad hoc teamwork (NAHT), it is unclear what the purpose of constructing a theoretical grounding is. A theoretical grounding for its own sake is deeply un-satisfying.

2. **Novelty**: Similar frameworks that decompose value functions according to the Shapley value have extensively been used in the literature with similarly strong experimental results.

3. **Practical Utility**: The paper does show better performance of applying the proposed Banzhaf (BM) and Shapley (SM) machines in two tasks: MPE and SMAC. However it is unclear what the computational overhead of the proposed methodologies is, thus, the practical tradeoffs of this method are not clearly stated. The paper also notes stability issues for SM on larger maps, an issue that is not investigated and might constitute a major limitation of SM.

Another minor issue is that the authors claim in Figure 2 that t the number of n-step return components in TTD($\lambda$) can be determined by the number of non-empty basis games. But this is not clear to me from interpreting the results: in 5v6 and 8v9, there are no clear differences between the performance of different values of $m$ and in MPE-pp it is actually the value of $m=14$ the one with highest performance.

**Questions:**

My main questions for the authors are as follows:

1. What was the orginial motivation for constructinn this axiomatic framework? What is one practical advantage of this framework over the existing heuristical methods?
2. When should a practitioner prefer SM vs BM, what are the properties of the task where one machine may be preferrable over the other?
3. What have been your observations about the stability of these methods in practice? In particular, related to SM, what seem to be the environments where the instability becomes apparent and what might be causing it?

---

> ### Author Response · Authors · 2025-11-21
>
> We appreciate you spending time on reviewing our paper and providing useful comments that help us clarify our work. We suppose that the main concerns you have are focused on the motivation and novelty of our axiomatic framework and the practicality of this framework. These are really important points for readers who are intended to understanding the work. We have tried our best to providing detailed answers. We sincerely hope these answers can reduce your concerns.
>
> ### Concerns
>
> **1. Problem motivation: The authors make no statement about the motivation of the problem or what they are attempting to achieve concretely. Although previous approaches rely on heuristics, and this paper claims to construct a theoretical grounding for ad hoc teamwork (NAHT), it is unclear what the purpose of constructing a theoretical grounding is. A theoretical grounding for its own sake is deeply un-satisfying.**
>
> We appreciate your perceptive thought and agree with your viewpoint, which is helpful to improve clarity of our paper. **To improve clarity of the motivation, we have refined the statement in the Introduction and Section 3 in the updated version of paper.**
>
>
> **2. Novelty: Similar frameworks that decompose value functions according to the Shapley value have extensively been used in the literature with similarly strong experimental results.**
>
> We appreciate your thought, but we **respectfully disagree with some of points**.
>
> We agree that the concept of Shapley values has been applied to cooperative MARL [1,2,3], but to our best knowledge **it has not been applied to the NAHT** yet. Note that cooperative MARL and NAHT are two **different problems**. In the cooperative MARL problem, **all** agents are controlled. In the NAHT problem, only **part** of agents are controlled. Also, there are key features of the NAHT that the common cooperative MARL does not possess: the varying number of controlled agents and the varying number of uncontrolled agents with various agent types [6].
>
> From the perspective of designing Shapley value based algorithms, previous work mainly focused on how to incorporate Shapley values into RL algorithms by shaping or fitting agent individual values via **matching the explicit formula** of Shapley values [1,2,3]. In contrast, in our paper we implement Shapley value based algorithms by **fulfilling the three axioms that uniquely determine it**. This is in principle a **different**, **novel** perspective to the previous work. The advantages of using axiomatic frameworks (our work) to incorporate Shapley values into RL algorithms are as follows:
>
> - Shapley values can be **generically** and **easily** incorporated into **any** RL algorithms (for the NAHT problem), only following the procedure we summarised in Section 4.6. This is a **novelty** that **cannot** be approached by the previous work that designed Shapley value based algorithms case by case.
>
> - It can deepen the understanding of Shapley values' axioms to investigate which axioms are beneficial for solving the NAHT problem. In this way, the useful axioms can be extracted and kept, rather than **completely** getting rid of the whole Shapley value approach. This is a benefit that **cannot** be approached by the previous work (that shapes or fits agent individual values via matching the explicit formula of Shapley values).

---

> ### Author Response · Authors · 2025-11-21
>
> **3. Practical Utility: The paper does show better performance of applying the proposed Banzhaf (BM) and Shapley (SM) machines in two tasks: MPE and SMAC. However it is unclear what the computational overhead of the proposed methodologies is, thus, the practical tradeoffs of this method are not clearly stated. The paper also notes stability issues for SM on larger maps, an issue that is not investigated and might constitute a major limitation of SM.**
>
> We appreciate your comments that can help clarify our work.
>
> **Answer to "However it is unclear what the computational overhead of the proposed methodologies is, thus, the practical tradeoffs of this method are not clearly stated":**
>
> - BM has not added extra computational overhead to the base RL algorithms that it is imposed on. Instead, the TTD($\lambda$) derived from the Linearity axiom only requires $M$ rewards to compute the target return, where $M$ is the number of non-empty coalitions. This is **more computationally efficient** than TD($\lambda$) that is applied in default to the base RL algorithms, when the number of agents is not too large (requiring the number of rewards equal to the episode length $T > M$).
>
>   We acknowledge that when the number of agents is increasing, $M > T$ could happen. However, this can be potentially resolved by additionally searching a partition of the team into several small subgroups, where each subgroup is with the number of agents that yields $M < T$. The computational overhead of the extra function to generate the partition will not be too high. It can be implemented in either a small-size network [4] or a non-parametric statistical method such as K-means [5] that have been attempted in the literature of MARL. **More importantly, the principle behind partitioning the team of agents is still consistent with the cooperative game theory that underpins our axiomatic framework.** We will investigate this direction in the future work.
>
> - Different from BM, SM only requires an extra reward shaping and an constraint on the sum of individual value functions. Both depend on **element-wise operations** that are implemented in **parallel computing** on GPUs, so there will be **no much computational overhead** incurred.
>
> **Answer to "The paper also notes stability issues for SM on larger maps, an issue that is not investigated and might constitute a major limitation of SM":**
>
> - We acknowledge that there could be some instability issue for the SM on larger maps, however, **we have investigated that in Appendix F.4 in the updated version of paper**. At the moment, we find that using value loss clip can mitigate the instability issue (see Figure 17). Nevertheless, we agree with you that it is required to find a better and more generic solution to this issue in the future work.
>
>
> **4. Another minor issue is that the authors claim in Figure 2 that t the number of n-step return components in TTD($\lambda$) can be determined by the number of non-empty basis games. But this is not clear to me from interpreting the results: in 5v6 and 8v9, there are no clear differences between the performance of different values of $m$ and in MPE-pp it is actually the value of $m=14$ the one with highest performance.**
>
> We appreciate your responsible attitude of promising the reasonableness of experimental results.
>
> In 8v9, the convergence rate of various $m$ has a pronounced deviation in our viewpoint. In 5v6, we agree that the performances between $m=20$ and $m=31$ are close. In MPE-PP, we acknowledge that the performance of $m=14$ is indeed a little bit higher than $m=7$. This could be due to some other confounding factors during learning, which we need to investigate in the future work. Nevertheless, in general the performances between these two $m$ values are close.
>
> Here, we may make a guess on the possible reason inferred from our axiomatic framework that may lead to the above phenomenon:
>
> - In Definition 4.3, we currently presume that the mapping between $V(C_i,s)$ and $E_\pi\big[G_{t:t+n_{i}}\mid s_t=s\big]$ follows $n_{i} = i$. However, it is possible that there exists a parametric function $n_{i} = f(i; \theta_{i})$, where $f(\cdot; \theta_{i})$ depends on environments or tasks. This implies that our current mapping $n_{i} = i$ could be only an **approximation** of the underlying exact mapping $f(\cdot; \theta_{i})$. The further investigation is left for the future work.
>
> The above case manifests that our axiomatic framework is worth further investigation in the future, which could help improve the algorithm design of the NAHT problem.

---

> ### Author Response · Authors · 2025-11-21
>
> ### Questions
>
> **1. What was the original motivation for construction this axiomatic framework? What is one practical advantage of this framework over the existing heuristic methods?**
>
> We appreciate this perceptive question to understand the gist and purpose of our work.
>
> **Motivation of establishing our axiomatic framework and its related practical advantages:**
>
> - The first motivation is deriving a new perspective of designing algorithms for the NAHT problem by considering its features (e.g. the different learning objectives from the common MARL) like what we did in Section 3. This can make the designing flow more transparent, which is the advantage that the heuristic method does not possess.
>
> - The second motivation is investigating whether the Shapley value's axioms can provide a principled foundation for solving the NAHT. This can leads to more efficient designing flow that the heuristic method cannot approach. For example, we can fast establish Banzhaf Machine by removing the Efficiency axiom from the Shapley Machine, with no need to establish this method from scratch again.
>
> - The final motivation is interpreting why TD($\lambda$) is useful for the heuristic methods POAM and IPPO-NAHT for the NAHT problem, so as to make it more trustworthy.
>
> **Additional practical advantage:**
>
> - In addition to the motivation-related practical advantages, an important consequence of our axiomatic framework is that the traditional $TTD(\lambda)$ has been generalised. In the traditional $TTD(\lambda)$, the weightings of n-step returns is determined by a geometric distribution governed by $\lambda$ (see Eq. (12)), which in our axiomatic framework is interpreted as complying with superadditive games. However, if considering game types other than superadditive games from the literature of cooperative game theory, the weightings are not necessarily governed by a geometric distribution and can be generalised to a broader function class (as we discussed in the Conclusion). **To motivate this future work, we move the NUMBER AND WEIGHTINGS OF BASIS GAME VALUES section from Appendices to Section 5.2 in the updated version of paper.** This opens a new door to explore more possibilities of **designing new RL algorithms in practice** for the NAHT problem, with game-theoretical interpretation by our axiomatic framework.
>
>
> **2. When should a practitioner prefer SM vs BM, what are the properties of the task where one machine may be preferable over the other?**
>
> We thank for this question that helps manifest the practicability of our methods.
>
> In our experiments, we find that the SM could be more preferable if the aim of task is for tackling unseen conventions. If the aim of task is for tackling unseen agent types, the BM could be more preferable. If the aim of task is handling in-distribution cases where the set of controlled agents is uncertain, the performance between SM and BM are similar. However, due to the computational overhead of BM is lower than that of SM, BM could be more preferable.
>
>
> **3. What have been your observations about the stability of these methods in practice? In particular, related to SM, what seem to be the environments where the instability becomes apparent and what might be causing it?**
>
> We appreciate this thoughtful question that helps clarify the feasibility of our methods.
>
> The key factor that causes the instability of the SM is highly likely the shaping reward that fulfils the Efficiency axiom (see Eq. (10)). We have justified this through the experimental result in **Appendix F.2 and Figure 13 in the updated version of paper**. Since the shaping reward is formed by value estimation, the essential reason for the instability is the **unstable value estimation** during learning. The unstable value estimation may appear due to the two following situations:
>
> - In the beginning of learning, the value estimation is not accurate enough due to the lack of data collected from the interaction with environments.
>
> - In the middle of learning, the value estimation could be with a sudden change (due to underestimation or overestimation), when uncontrolled teammates of unseen types are met.
>
> To mitigate this instability, we add a multiplier $\alpha$ in practice to control the extent of the difference between two value estimation terms in the shaping reward (see Eq. (22) in Appendix E.1.1).

---

> > ### Author Response · Authors · 2025-11-21
> >
> > ### References
> >
> > [1] Wang, Jianhong, Yuan Zhang, Tae-Kyun Kim, and Yunjie Gu. "Shapley Q-value: A local reward approach to solve global reward games." In _Proceedings of the AAAI conference on artificial intelligence_, vol. 34, no. 05, pp. 7285-7292. 2020.
> >
> > [2] Han, Dongge, Chris Xiaoxuan Lu, Tomasz Michalak, and Michael Wooldridge. "Multiagent model-based credit assignment for continuous control." arXiv preprint arXiv:2112.13937 (2021).
> >
> > [3] Wang, Jianhong, Yuan Zhang, Yunjie Gu, and Tae-Kyun Kim. "Shaq: Incorporating shapley value theory into multi-agent q-learning." _Advances in Neural Information Processing Systems_ 35 (2022): 5941-5954.
> >
> > [4] Ma, Jinming, and Feng Wu. "Feudal multi-agent reinforcement learning with adaptive network partition for traffic signal control." _arXiv preprint arXiv:2205.13836_ (2022).
> >
> > [5] Müller, R., Turalic, H., Phan, T., Kölle, M., Nüßlein, J., & Linnhoff-Popien, C. (2024). ClusterComm: Discrete communication in decentralized marl using internal representation clustering. _arXiv preprint arXiv:2401.03504_.
> >
> > [6] Wang, Caroline, Muhammad Arrasy Rahman, Ishan Durugkar, Elad Liebman, and Peter Stone. "N-agent ad hoc teamwork." Advances in Neural Information Processing Systems 37 (2024): 111832-111862.

---

### Official Review · Reviewer_m7D6 · 2025-10-29

**Soundness:** 3
**Presentation:** 3
**Contribution:** 2
**Rating:** 4
**Confidence:** 4

**Summary:**

This paper introduces an **axiomatic framework for N-Agent Ad Hoc Teamwork (NAHT)** grounded in Shapley’s cooperative game theory. By reinterpreting three axioms as structural constraints on NAHT learning, the authors derive two theoretically principled algorithms, Shapley Machine (SM) and Banzhaf Machine(BM), which were evaluated on MPE-PP and SMAC in NAHT setting. The results demonstrate that these methods perform well. Furthermore, the experiments indicate that relaxing the Efficiency axiom can even lead to better agent type generalization than enforcing the full set of Shapley axioms. But the proposed methods do not explicitly address the challenges specific for NAHT.

**Strengths:**

1.  The paper achieves a theoretically rigorous fusion of NAHT and cooperative game theory, providing a clear axiomatic basis for agent value decomposition.
2. The experiments are well designed and cover multiple NAHT scenarios, demonstrating the practical feasibility and stability of both SM and BM.
3. The framework is internally self-consistent, with well-motivated propositions that link the axioms to learning method in a clean way.

**Weaknesses:**

1.  The formal axioms rely on strong assumptions such as additive reward decomposition and state-wise independence, which are unlikely to strictly hold in real NAHT environments.
2. Despite extensive proofs, many theoretical results merely reinterpret Shapley axioms to constrain the learning structure, without introducing fundamentally new optimization principles specific to NAHT. The proposed methods do not explicitly address the defining challenges of NAHT, such as teammate uncertainty, heterogeneity, adaptive coordination and so on. Thus limiting the framework’s task-specific effectiveness.

**Questions:**

**Q1.** How does the proposed axiomatic framework ensure robustness when agents operate in highly uncertain or partially observable environments? Since the axioms (Efficiency, Symmetry, Linearity) are defined under idealized cooperative settings, it remains unclear how the framework maintains stable performance when the underlying dynamics or teammate behaviors are stochastic or adversarial.

**Q2.** Could the authors provide a more intuitive interpretation of how the proposed axiomatic structure concretely addresses the specific challenges of NAHT? A higher-level conceptual or visual explanation could help clarify why enforcing Shapley-inspired constraints improves the perfoemance specific in NAHT setting, rather than a generic framework suitable for both AHT and MARL.

---

> ### Author Response · Authors · 2025-11-21
>
> We appreciate your time on providing us with such insightful comments, which help us re-examine the value of our work. We suppose that the main concerns you have are focused on the necessity of our axiomatic framework and how the resulting techniques of the framework related to the specific setting of the NAHT. We have tried our best to clarify these points. We sincerely hope our answers can reduce your concerns.
>
> ### Concerns
>
> **1. The formal axioms rely on strong assumptions such as additive reward decomposition and state-wise independence, which are unlikely to strictly hold in real NAHT environments.**
>
> We appreciate your insightful thought, but we are afraid that there could be some misunderstanding.
>
> We agree with you that the Efficiency axiom (additive reward decomposition) **may not be certainly suitable for** the NAHT problem at the moment. We have verified this through experiments for the generalisation to unseen agent types, where the Efficiency axiom could weaken the performance compared with our algorithm only with other two axioms (Banzhaf Machine). Note that the algorithm with the Efficiency axiom (Shapley Machine) still performs better than the baseline algorithms (without our axiomatic framework) in most of cases. The main purpose of mentioning the Efficient axiom here is for the completeness of our axiomatic framework. In practice, people can freely choose which axiom to impose on the base algorithm. On the other hand, it is hard to predict if the Efficiency axiom could help in the future, which is another reason why we leave this option in our axiomatic framework.
>
> As for the state-wise independence, **there exists some misunderstanding** possibly due to that our original statement of the Game-Structured NAHT is not clear enough. **We have refined the words of Definition 3.2 and corresponding explanation in Section 3.** We hope this can clarify things.
>
> Generally speaking, we apply state values as a structure to represent a trajectory of one NAHT process, so **states are dependent to each other**:
>
> - Suppose that a sub-trajectory denoted by "B" with a starting state belonging to the set of future states related to a trajectory "A" with the starting state of the NHAT process. In this case, the two trajectories "A" and "B" can be represented as two consecutive state values with the state representing their corresponding starting state. As a result, the two states are dependent that is governed by the Markov dynamics, and their corresponding state values can be associated by the Bellman equation, like what we show in Eq. (4).
>
>
> **2. Despite extensive proofs, many theoretical results merely reinterpret Shapley axioms to constrain the learning structure, without introducing fundamentally new optimization principles specific to NAHT. The proposed methods do not explicitly address the defining challenges of NAHT, such as teammate uncertainty, heterogeneity, adaptive coordination and so on. Thus limiting the framework’s task-specific effectiveness.**
>
> We agree with you that our paper did not follow the conventional route on solving the NAHT problem (by either developing specific optimisation techniques or targeting any subproblem). However, **we would like to respectfully argue that this is not a weakness**. Rather, our work brings new ideas designed for the features of the NAHT problem that can be applied to **any base algorithms**. Furthermore, the techniques derived by our axiomatic framework can be combined with other specific techniques for solving any subproblem you mentioned.
>
> For example, our axiomatic framework derives that TTD($\lambda$) is more suitable for the NAHT problem than the vanilla TD($\lambda$) inherited from the IPPO algorithmic setting. **This result cannot be discovered without our axiomatic framework.** To clarify the relation between TTD($\lambda$) and the NAHT, we move the NUMBER AND WEIGHTINGS OF BASIS GAME VALUES section from Appendices to Section 5.2 in the updated version of paper. Furthermore, the agent modelling technique is an implementation of the Symmetry axiom, providing a consistent framework for understanding this technique. Note that the Symmetry axiom is related to the heterogeneity you mentioned (see the answer to Question 1). **This consistency of our axiomatic framework may potentially lead to new ideas and avoid confusing technique development for the NAHT problem.** Moreover, the refined version of Section 3 and the Introduction section in the updated version of paper (highlighted in red) also emphasise how our axiomatic framework is specific for the NAHT.

---

> ### Author Response · Authors · 2025-11-21
>
> ### Questions
>
> **1. How does the proposed axiomatic framework ensure robustness when agents operate in highly uncertain or partially observable environments? Since the axioms (Efficiency, Symmetry, Linearity) are defined under idealized cooperative settings, it remains unclear how the framework maintains stable performance when the underlying dynamics or teammate behaviors are stochastic or adversarial.**
>
> We appreciate your good question that dives into the detail of our framework.
>
> Whether environments are partially observable or not, is **orthogonal** to the three axioms we proposed, so addressing partial observability is **out of the scope** of our axiomatic framework's functionality. Since we consider our axiomatical framework building on Dec-POMDP, any technique designed for partial observability can be applied in combination with the techniques derived from our axiomatic framework, e.g., the policies of our BM- and SM-frameworks are implemented in RNNs.
>
> As for the highly uncertainties of agents, they are mainly from two aspects: (1) the varying number of controlled agents, and (2) the varying number and types of uncontrolled agents [1]. **This has been taken into consideration when constructing the representation of state-specific cooperative game values for the NAHT problem in Section 3.** We have improved the description to emphasise this point in the updated version of paper. The summary is as follows:
>
> - The foundation for all the three axioms appearing in Theorem 4.10 is the state-specific cooperative game space, as defined in Definition 3.2. This game space is specifically designed for the NAHT problem, to consider two learning sub-objectives **specific for the NAHT**: (1) Learning internal cooperation among **the varying number of controlled agents**; (2) Learning external teamwork with **the varying number and types of uncontrolled agents**. Briefly, a state-specific cooperative game value can be decomposed as follows:
>   $$
>    v\_{s} = \underbrace{\sum\_{C\_{int} \in \mathcal{P}\_{+}(\mathcal{C})} k\_{C\_{int}} v\_{C\_{int},s}^{1}}\_{\textbf{Internal Cooperation}} + \underbrace{\sum\_{C\_{ext} \in \mathcal{P}\_{+}(\mathcal{M}) \setminus \mathcal{P}\_{+}(\mathcal{C})} k\_{C\_{ext}} v\_{C_{ext},s}^{1}}\_{\textbf{External Teamwork}}.
>    $$
>
>    The value structure above is analogous to setting up curricula for facing uncertain number of controlled and uncontrolled agents. Since our axiomatic framework is built on this game space, this makes it naturally endowed with the capability of dealing with uncertainties emerging in these two learning sub-objectives.
>
> The stochastic underlying dynamics of teammate (uncontrolled agent) behaviours are addressed by the Symmetry axiom. Under our axiomatic framework, the Symmetry axiom inherited from the cooperative game theory is the key to differentiating heterogeneous agents. **Identifying those teammates under stochastic behaviours is the key to keeping robustness of the controlled agents' performance** for the NAHT problem. In practice, this can be implemented with the agent modelling techniques, which we have proved to fulfil the basic requirement of the Symmetry axiom (see Section 4.5 and Theorem 4.8).
>
> About the adversarial behaviours of agents, to be honest we have not considered it in our axiomatic framework at the moment. **To our best knowledge, this is not a common subproblem considered in the NAHT problem.** However, we agree that it is worth investigating in the future work to enable the ad hoc team more robust when encountering sabotage from teammates.

---

> ### Author Response · Authors · 2025-11-21
>
> **2. Could the authors provide a more intuitive interpretation of how the proposed axiomatic structure concretely addresses the specific challenges of NAHT? A higher-level conceptual or visual explanation could help clarify why enforcing Shapley-inspired constraints improves the performance specific in NAHT setting, rather than a generic framework suitable for both AHT and MARL.**
>
> We appreciate your good question that can help clarify our work.
>
> Since we have mentioned some of intuitive interpretation in the answer to Question 1, we will write a summary to conclude those points, as follows:
>
> - The construction of state-specific cooperative game space in Section 3 is **specific for** the NAHT problem, which considers the learning sub-objectives distinct from the standard MARL, caused by the varying number of controlled agents and the varying number of uncontrolled agents with various agent types during learning (all those are key features of the NAHT problem). Our axiomatic framework is built on this game space.
>
> - The **Linearity axiom** is closely associated with this construction that is for deriving the RL technique for the NAHT problem, such as TTD($\lambda$). We have shown that this RL technique can on average improve the performance of the NAHT (e.g., BM-POAM vs. POAM), especially in evaluation on unseen agent types and conventions.
>
> - The **Symmetry axiom** is corresponding to differentiating stochastic behaviours of different agents that may appear during learning, as we explained in the answer to Question 1.
>
> - The **Efficiency axiom** is for deriving constraints that help reduce the search space of individual values under stochastic uncontrolled agent behaviours during learning in practice.
>
> As for whether our framework can work more broadly to MARL, theoretically it **can** but **not necessarily** (with no motivation). Recall that the cooperative game value is formed by basis game values embedding situations involving the key features of the NAHT problem: the varying number of controlled agents and the varying number of uncontrolled agents with various agent-types. In contrast, the standard MARL only requires learning for **a fixed number of controlled agents** (with no consideration of any uncontrolled agent). This implies that the standard MARL does not necessarily require any complex structure like the state-specific cooperative game space we establish for the NAHT, not even the corresponding axiomatic framework.
>
> As for whether our framework can work more broadly to AHT, yes it can. The reason is direct and natural: the AHT problem is a **special case** of the NAHT problem, and any approach for the NAHT problem in principle can be applied to the AHT problem by fixing N = 1.
>
>
> ### References
>
> [1] Wang, Caroline, Muhammad Arrasy Rahman, Ishan Durugkar, Elad Liebman, and Peter Stone. "N-agent ad hoc teamwork." Advances in Neural Information Processing Systems 37 (2024): 111832-111862.

---

### Official Review · Reviewer_UQNF · 2025-10-30

**Soundness:** 2
**Presentation:** 2
**Contribution:** 2
**Rating:** 2
**Confidence:** 3

**Summary:**

The paper proposes an algorithmic framework for a recently introduced multi-agent systems model of n-player ad hoc teamwork. The main results are Theorem 4.8 (a sufficient condition for resulting value functions) and Theorem 4.10 (saying that a particular algorithm recovers Shapley axioms). Furthermore, the paper presents simulations.

**Strengths:**

The paper works on building algorithms that construct cooperative game theory concepts in their algorithms, which is an interesting question.

**Weaknesses:**

1. The theory does (for my taste) not sufficiently motivate why the paper considers the questions it considers in the NAHT setting. This can be seen in the proof of one of the main results, Theorem 4.10, which relies on reducing the game's properties to the state conditioned game's properties. Why does the method not work more broadly?
2. The writing of the paper is not clear enough yet for publication in a top venue. Proofs are informal. The important role of reduction of the state-conditioned game makes me believe that the paper is not using the appropriate scope for the generality of its method.
3. Many of the experiments don't reach statistical significance. Some are hard to read. For example, Figures 2 and 3, which are the main training curves, are too small for me to evaluate which methods perform best.

**Questions:**

- How is your analysis specific to NAHT? Can your method be applied more generally?
- Which of your simulation results are statistically significant?

---

> ### Author Response · Authors · 2025-11-21
>
> We appreciate your insightful comments on our paper, which gives us opportunities to clarify our work. We suppose that the main concerns from you are the relation between our theory and the motivating questions, as well as the statistical significance. We have tried our best to addressing your concerns, which we believe are mainly caused by some misalignment of concepts between us. We sincerely hope your concerns can be reduced after reading our rebuttal.
>
> ### Concerns:
>
> **1. The theory does (for my taste) not sufficiently motivate why the paper considers the questions it considers in the NAHT setting. This can be seen in the proof of one of the main results, Theorem 4.10, which relies on reducing the game's properties to the state conditioned game's properties. Why does the method not work more broadly?**
>
> We agree that you are almost right for the workflow of technical detail.
>
> We guess that you may imply that **how the research questions we raise in the Introduction are aimed at motivating the establishment of our theory (the axiomatic framework) that improves the algorithm design of the NAHT**. To emphasise the motivation for our axiomatic framework for the NAHT, we **improve our words of research questions in the Introduction** and **clarify the link between the foundation of our axiomatic framework (GAME-STRUCTURED N-AGENT AD HOC TEAMWORK) and the NAHT in Section 3** in the updated version of paper.
>
> The summary of the refined research questions are as follows:
>
> - The existing method for the NAHT is designed heuristically **without** rigorous theoretical grounding, which undermines trustworthiness.
> - The fundamental distinction between the learning objectives of NAHT and MARL was **overlooked** when designing the the method.
> - The existing method employs TD($\lambda$) to train its critics, but its relation to the NAHT objective remains **unclear**.
>
> The relation between our axiomatic framework and the first and third research questions are intuitive:
>
> - Our axiomatic framework provides a theoretical framework to improve algorithm design with trustworthiness.
> - Based on our axiomatic framework, we can derive that the Linearity axiom can lead to the TTD($\lambda$), which gives a direct reason why TD($\lambda$) is useful to some extent.
>
> The relation between our axiomatic framework and the second research question is as follows:
>
> - The foundation for all the three axioms appearing in Theorem 4.10 is the state-specific cooperative game space, as defined in Definition 3.2. This game space is specifically designed for the NAHT problem, to consider two learning sub-objectives **specific for the NAHT**: (1) Learning internal cooperation among **the varying number of controlled agents**; (2) Learning external teamwork with **the varying number and types of uncontrolled agents**. Briefly, a state-specific cooperative game value can be decomposed as follows:
>   $$
>    v\_{s} = \underbrace{\sum\_{C\_{int} \in \mathcal{P}\_{+}(\mathcal{C})} k\_{C\_{int}} v\_{C\_{int},s}^{1}}\_{\textbf{Internal Cooperation}} + \underbrace{\sum\_{C\_{ext} \in \mathcal{P}\_{+}(\mathcal{M}) \setminus \mathcal{P}\_{+}(\mathcal{C})} k\_{C\_{ext}} v\_{C\_{ext},s}^{1}}\_{\textbf{External Teamwork}}.
>    $$
>
>   The value structure above is analogous to setting up curricula for facing uncertain number of controlled and uncontrolled agents. Since our axiomatic framework is built on this game space, this makes the axiomatic framework **closely related to the NAHT, rather than a general framework with no motivation or scope**.
>
> - In contrast, the standard MARL **only** requires learning to establish internal cooperation among **a fixed number of controlled agents** (with no consideration of any uncontrolled agent). This implies that the standard MARL does not necessarily require any complex structure like the state-specific cooperative game value we establish for the NAHT.

---

> ### Author Response · Authors · 2025-11-21
>
> **2. The writing of the paper is not clear enough yet for publication in a top venue. Proofs are informal. The important role of reduction of the state-conditioned game makes me believe that the paper is not using the appropriate scope for the generality of its method.**
>
> We are sorry to hear that the writing of the paper is unclear to you. We hope the answer to Concern 1 above can help clarify things.
>
> About how to identify whether a proof is formal or not, we may have different ideas. All results that we have proved are building on **explicit conditions/assumptions** with logical procedures of proof, which in our viewpoint is **appropriately formal**. For this reason, we **respectfully disagree** with your statement that proofs are informal, but we **respect** your own taste on evaluating what formal proofs are.
>
> Concerning your comment "The important role of reduction of the state-conditioned game makes me believe that the paper is not using the appropriate scope for the generality of its method", we are afraid that **there could be some misunderstanding**. We hope our answer to Concern 1 and the updated version of paper can clarify this point.
>
> **3. Many of the experiments don't reach statistical significance. Some are hard to read. For example, Figures 2 and 3, which are the main training curves, are too small for me to evaluate which methods perform best.**
>
> We apologise for the small figures due to the limited space of 9 pages. In Figure 3, it is a fact that some of our methods perform close to each other, but all of them perform statistically significantly better than the baselines. In Figure 2, the $m$ values obtained by our theoretical framework can reach the nearly optimal performance. Thanks to the one extra page allowed in the stage of rebuttal, we have enlarged the size of Figure 2 in the updated version of paper.
>
> About the statistical significance, we have reported all **genuine** results that are obtained by evaluating each algorithm with **5 random seeds** governing randomness of training procedures and **128 random trials** of interaction with the environment for each random seed, **as what has been done in the previous work that proposed the NAHT** [1].
>
> We **agree** that some of these results' 95% confidence intervals (CIs) could be with excessive overlapping due to the wide range of 95% CIs, e.g. out-of-distribution teammates and unseen convention evaluations (Figures 4 and 8). However, the results of out-of-distribution evaluations consider the  performance by averaging **all pairings of N controlled and M − N uncontrolled agents and their corresponding random seeds  (M-N score)** as consistent with the previous work [1], as we have clarified in the caption of Figure 4. For more detail about this M-N score, please refer to Appendix E.3. Accordingly, the overlapping is caused by **the variation of performances across all pairings of N controlled and M − N uncontrolled agents, rather than the randomness of performance with no statistical significance as you mentioned**. To further clarify this, we have reported the new results across each specific number of controlled agents in Appendix F.5 in the updated version of paper.
>
> ### Questions
>
> **1. How is your analysis specific to NAHT? Can your method be applied more generally?**
>
> We appreciate your insightful question that is the key to understanding our work. The short answer to the specificity to the NAHT is that **our state-specific cooperative game model has considered the learning objective specific for the NAHT problem**. The short answer to whether our framework can be applied to the standard MARL is that in principle it **can** but **not necessary** (with no motivation). We have explained the reasons in detail in the answer to Concern 1.
>
>
> **2. Which of your simulation results are statistically significant?**
>
> We believe this issue can been addressed by **the new provided results in Appendix F.5**, as we mentioned in the answer to Concern 3 above. For other experimental results, most of them are statistically significant (with half-arm overlapping of CIs) to show the better performance of (at least one of) our methods than the baselines. In a rough calculation, in 85% of situations at least one of our algorithms are with statistically significantly better performance than the baselines.
>
> However, we **agree** with you that **not** all of them are **highly statistically significant** (with no overlapping between CIs). We **respectfully** argue that this is mainly led by the difficulty of the NAHT problem. As a result, **this is not a critical issue that is enough to hide the strength and novelty of our work**.
>
>
> ### References
>
> [1] Wang, Caroline, Muhammad Arrasy Rahman, Ishan Durugkar, Elad Liebman, and Peter Stone. "N-agent ad hoc teamwork." Advances in Neural Information Processing Systems 37 (2024): 111832-111862.

---

### Official Review · Reviewer_cBT1 · 2025-11-01

**Soundness:** 3
**Presentation:** 3
**Contribution:** 3
**Rating:** 6
**Confidence:** 2

**Summary:**

The paper proposes an axiomatic view of n‑agent ad hoc teamwork (NAHT). Instead of plugging the closed‑form Shapley value into MARL, it models each state as a cooperative game and enforces Shapley’s axioms—Linearity, Symmetry, Efficiency—as structural constraints on per‑agent value functions. This yields two learning templates: Shapley Machine (enforces all three axioms) and Banzhaf Machine (drops Efficiency, aligning with the Banzhaf index). A key insight is that Linearity maps directly to truncated TD(λ) with positive weights over n‑step returns; the number of return components is tied to the number of non‑empty coalitions. Built on IPPO and POAM, these variants generally improve training performance; notably, relaxing Efficiency can generalize better to unseen teammate types. Experiments on MPE‑PP and SMAC support these claims.

**Strengths:**

1. Recasting Shapley’s axioms as trainable constraints gives a principled design space: enforcing all axioms recovers Shapley; dropping Efficiency yields Banzhaf. This connects cooperative game theory to critic shaping in a way that’s easy to implement.
1. The paper shows Linearity to truncated TD(λ) with strictly positive weights, offering a game‑theoretic reading of λ‑return weighting. This is both novel and practically actionable.
1. Comprehensive experimental design.
1. Clear training losses ablations on m and λ, and full hyperparameters make the work easy to replicate.

**Weaknesses:**

1. Several technical steps rely on superadditivity and variance/auto‑covariance assumptions (e.g., Lemma C.1, Assumption C.2) whose realism in complex NAHT environments is not empirically validated.
1. Implementing Efficiency requires accurate per‑agent value estimates for uncontrolled teammates, approximated via shared critics and a tunable coefficient α to stabilize learning; this may introduce bias and add a non‑axiomatic knob.
1. Evaluations focus on MPE‑PP/SMAC; no continuous‑control or real‑robot domains and limited comparison to the most recent open‑team frameworks beyond IPPO/POAM baselines—so external validity is still uncertain.

**Questions:**

How do you handle the uncertainty/bias in value estimation, especially for uncontrolled teammates?

---

> ### Author Response · Authors · 2025-11-21
>
> We appreciate you spending time in reviewing our paper. All of your comments are useful for us to better clarify our work in detail. The main concerns you have concentrate on whether some of conditions are aligned with the NAHT setting (by empirical assessment) and how we deal with the uncertainty of value estimation. We draft answers to those concerns. We sincerely hope our answers can reduce you concerns.
>
> ### Concerns:
>
> **1. Several technical steps rely on superadditivity and variance/auto‑covariance assumptions (e.g., Lemma C.1, Assumption C.2) whose realism in complex NAHT environments is not empirically validated.**
>
> We understand and **respect** your spirit in evaluating techniques with empirical experiments.
>
> The superadditivity is a sufficient condition leading to team reward games (the superset of Dec-PODMPs) that has been shown in [1] and Footnote 5 in our paper, which is the foundation for the NAHT problem to specify that all agents own a shared goal (since the NAHT is described as the Dec-POMDP [2]). As a result, it is a fact of the NAHT setting that does not necessarily require experimental verification.
>
> Assumption C.2 is derived by a mathematical reasoning based on the Game-Structured NAHT defined in Definition 3.2, as explained in Remark C.3. As a result, it is a mathematical fact based on our theoretical framework. However, we **agree** with you that this is not necessarily a fact the aligns with the NAHT.
>
> Although we admit that it is difficult for us to observe the **direct phenomenon** for Assumption C.2 at the moment, you may think of a workaround in such way:
>
> - Assumption C.2 is derived by Definition 3.2, and both of them are the foundations to derive our practical methods SM and BM. This implies that if we can show the effectiveness of SM and BM in the NAHT environment, then we empirically verify the reasonableness of Assumption C.2 in a indirect way.
>
> By the above principle, we have **indirectly verified the reasonableness of Assumption C.2** and the validity of our theoretical framework. Nevertheless, we **agree** with you that observing direct phenomenon of Assumption C.2 is more preferable, and we will approach that in the future work.
>
>
> **2. Implementing Efficiency requires accurate per‑agent value estimates for uncontrolled teammates, approximated via shared critics and a tunable coefficient α to stabilize learning; this may introduce bias and add a non‑axiomatic knob.**
>
> We **agree** with your viewpoint, but **we would like to respectfully emphasise that this is unavoidable at the moment due to the nature of the NAHT problem**. Since controlled agents do not have any knowledge about the number or types potential uncontrolled teammates, it is impossible to structure explicit critics for each of them. This is the reason why we implement the Efficiency axiom with this strategy. We will investigate better solutions to this issue in the future work.
>
> On the other hand, the non-axiomatic technique related to the implementation is not conflicting with the axiomatic framework. Instead, **they can co-exist**, where the axioms play the role of guiding the "direction", while the non-axiomatic technique is responsible for specifying the axioms to make them fit the complicated situation.
>
>
> **3. Evaluations focus on MPE‑PP/SMAC; no continuous‑control or real‑robot domains and limited comparison to the most recent open‑team frameworks beyond IPPO/POAM baselines—so external validity is still uncertain.**
>
> We **agree** with your spirit of evaluating a paper from a general view. However, we would like to **respectfully argue** that:
>
> - The NAHT problem **was proposed last year**, so it does not have so many baselines or environments to evaluate;
>
> - We have verified the effectiveness of our frameworks **on all effective algorithms and environments provided in the NAHT paper** [2] to our best effort;
>
> - Design a new environment for the NAHT is demanding, which requires rigorous validation of generated teammates [2].
>
> In the future work, we plan to design extra environments and evaluating our frameworks on more algorithms, following your suggestion.

---

> ### Author Response · Authors · 2025-11-21
>
> ### Questions:
>
> **1. How do you handle the uncertainty/bias in value estimation, especially for uncontrolled teammates?**
>
> We appreciate your insightful question.
>
> As we observed in experiments, the uncertainty of value estimation primarily appears in the beginning of training when the value estimation is not informed enough and in the middle of training when new types of uncontrolled teammates are met.
>
> We have not controlled this proactively, **unless** we apply the Efficiency axiom. The reason is that without proactive control for uncertainty/bias in value estimation, the BM approach (with no Efficiency axiom) learns well with no excessive uncertainty (the variance of value estimation is associated with that of the critic loss), as shown in Figure 7.
>
> In contrast, the Efficiency axiom relies on the shaping reward based on the value estimation, i.e., $R_{t,i} = R_{t} - \sum_{j \neq i} \left( V_{j}(s_{t}) - \gamma V_{j}(s_{t+1}) \right)$ (see Eq. (10)), so the uncertainty of value estimation may seriously affect the training process of value functions recursively, leading to "excessive biases" like the snowball effect. To mitigate this issue, our strategy is adding an multiplier $\alpha$ to control the uncertainties of value estimation in the shaping reward in practice, i.e., $R_{t,i} = R_{t} - \alpha \sum_{j \neq i} \left( V_{j}(h_{t,j}) - \gamma V_{j}(h_{t+1,j}) \right)$ (see Eq.(22) in Appendix E.1.1). With this remedy, we show in Figure 7 that the bad effect of the uncertainty of value estimation for the SM approach is removed.
>
> ### References
>
> [1] Wang, Jianhong, Yuan Zhang, Tae-Kyun Kim, and Yunjie Gu. "Shapley Q-value: A local reward approach to solve global reward games." In Proceedings of the AAAI conference on artificial intelligence, vol. 34, no. 05, pp. 7285-7292. 2020.
>
> [2] Wang, Caroline, Muhammad Arrasy Rahman, Ishan Durugkar, Elad Liebman, and Peter Stone. "N-agent ad hoc teamwork." Advances in Neural Information Processing Systems 37 (2024): 111832-111862.

---

### Author Response · Authors · 2025-11-21
**Overall Response**

We appreciate all reviewers for acknowledging our paper's contribution and giving us an opportunity for further clarifying and improving our work. According to reviewers' comments, we have improved our paper in the following ways:

- Improve the motivation of our work in the INTRODUCTION section, especially highlighting its link to the NAHT problem. (Reviewers yG89 and UQNF)

- Clarify the foundation of our axiomatic framework in Section 3: GAME-STRUCTURED N-AGENT AD HOC TEAMWORK, highlighting that its scope is related to the NAHT problem. (Reviewers UQNF and m7D6)

- Add a new Appendix F.5 section in the updated version of paper, clarifying the misunderstanding on the statistical significance of our experimental results of OOD evaluation with more detail. (Reviewer UQNF)

- Move the NUMBER AND WEIGHTINGS OF BASIS GAME VALUES section from Appendices to Section 5.2 in the main body of paper, clarifying the **potential value** of our axiomatic framework that can shed light on **generalising TD($\boldsymbol{\lambda}$)-related RL algorithms for the NAHT problem**. (Reviewer m7D6)

- Clarify the evaluation metrics in Appendix E.3.

All key modifications are highlighted in red in the updated version of paper.

We sincerely hope the modification of paper and the answers to your concerns can clarify the motivations, contributions and potential value of our paper.

---

### Author Response · Authors · 2025-12-04
**Summary of Our Paper and Reviewers' Concerns**

Dear Area Chair,

We would like to sincerely appreciate your extra contribution to the community during this special time. To ease your work, we now provide a brief summary of our work and reviewers' main concerns:

**Summary of Our Paper:**

This paper primarily proposes an axiomatic framework to address the N-Agent Ad Hoc Teamwork (NAHT). We first introduce the state-specific cooperative game space to model the process of NAHT, referred to as the game-structured NAHT. Then, we build on the axioms of the game-strcutured NAHT to structure agents' individual values, which can flexibly recover Shapley values and Banzhaf index on any base RL algorithms. More importantly, we also derive the TTD($\lambda$) from the Linearity axiom of our axiomatic framework. In experiments, we not only verify the reasonableness of the state-specific cooperative game space and show that the performance of algorithms with our axiomatic framework improves on the base algorithms, but also validate that each axiom has its particular effect on a dimension of the NAHT (unseen conventions or unseen agent types). As a result, people can feel free to select the axiom of interest due to the flexibility of deployment on any RL algorithms. Moreover, based on our axiomatic framework the representation of TTD($\lambda$) can be generalised to a broader class of functions with a theoretical interpretation.

**Summary of Main Concerns from Reviewers:**

- Reviewer cBT1: Whether some of conditions are aligned with the NAHT setting (by empirical assessment) and how we deal with the uncertainty of value estimation.
- Reviewer UQNF: The relation between our theory and the motivating questions, as well as the statistical significance of our experimental results.
- Reviewer m7D6: The necessity of our axiomatic framework and how the resulting techniques of the framework related to the specific setting of the NAHT.
- Reviewer yG89: The motivation and novelty of our axiomatic framework and the practicality of this framework.

We have tried our best to address all those concerns and clarified the possible misunderstanding from reviewers in our reponses and the updated version of paper. We appreciate your effort again on fairly evaluating our paper.

---

### Meta-Review · Area_Chair_zF8h · 2026-01-06

**Summary:**

This paper casts the n-agent ad-hoc teamwork (NAHT) problem as a state-specific cooperative game and shows how each of the axioms of Shapley values maps onto structural constraints for RL (e.g., truncated TD and Linearity axiom). Some of the reviewers appreciated these structural connections, however, the group had concerns about whether certain assumptions hold (e.g., auto-covariance), the motivation for the theoretical grounding, the specific tailoring to the NAHT setting, lack of statistically significant improved performance in some cases, and expressed a desire for additional experimental results / comparisons.

**Reviewer Concerns:**

The authors addressed most of the reviewers' concerns in the rebuttal and included a summary in their rebuttal summary. The most important outstanding concern is likely the motivation and specificity of the approach to the NAHT setting. I do not believe that the reviewers concerns regarding motivation have been fully allayed.

**Reviewer Scores:**

- cBT1: I think they likely would have kept their score at a 6.
- UQNF: It's possible they would have increased their score, but not by much. Maybe to a 3.
- m7D6: It's possible they would have increased their score, but not by much. Maybe to a 5.
- yG89: I think they likely would have kept their score at a 4.

---

### Decision · Program_Chairs · 2026-01-26

Reject